# Image as First-Order Norm+Linear Autoregression: Unveiling Mathematical Invariance

## Abstract

This paper introduces a novel mathematical property applicable to diverse images, referred to as FINOLA (First-Order Norm+Linear Autoregressive). FINOLA represents each image in the latent space as a first-order autoregressive process, in which each regression step simply applies a shared linear model on the normalized value of its immediate neighbor. This intriguing property reveals a mathematical invariance that transcends individual images. Expanding from image grids to continuous coordinates, we unveil the presence of two underlying partial differential equations. We validate the FINOLA property from two distinct angles: image reconstruction and self-supervised learning. Firstly, we demonstrate the ability of FINOLA to auto-regress up to a 256×256 feature map (the same resolution to the image) from a single vector placed at the center, successfully reconstructing the original image by only using three 3×3 convolution layers as decoder. Secondly, we leverage FINOLA for self-supervised learning by employing a simple masked prediction approach. Encoding a single unmasked quadrant block, we autoregressively predict the surrounding masked region. Remarkably, this pre-trained representation proves highly effective in image classification and object detection tasks, even when integrated into lightweight networks, all without the need for extensive fine-tuning. The code will be made publicly available.

## 1 Introduction

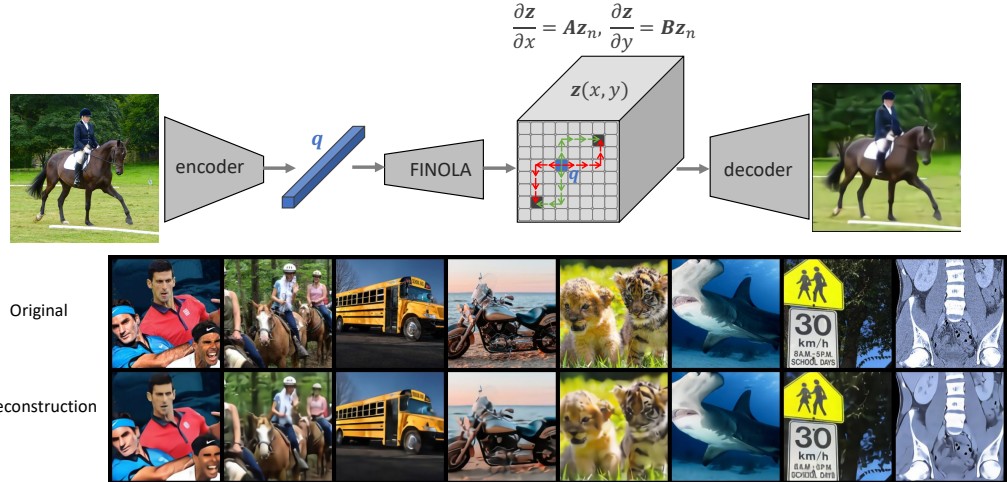

Figure 1: **FINOLA on image reconstruction.** Each image is firstly encoded into a single vector $q$. Then, FINOLA is applied to $q$ to iteratively generate the feature map $z(x, y)$ through autoregression, which is governed by two partial differential equations (PDEs). Finally, a decoder composed of upsampling and convolutional layers is used to reconstruct the image. Best viewed in color.

Autoregressive language models, as exemplified by GPT Radford et al. (2018; 2019); Brown et al. (2020), have achieved remarkable success in the realm of Natural Language Processing (NLP). These models generate text by predicting the probability distribution of the next word in a sequence,

Figure 2: **Masked FINOLA on self-supervised pre-training:** a single *unmasked* quadrant passes through encoder-FINOLA-decoder network to predict the surrounding *masked* region. Introducing masked prediction into FINOLA trades restoration accuracy for gaining semantic representation.

based on the preceding words. This success has not been confined to NLP; it has extended into Computer Vision, witnessed in the form of innovations like iGPT Chen et al. (2020a) for unsupervised learning, PixelCNN van den Oord et al. (2016a); Salimans et al. (2017) for image generation, and DALL-E Ramesh et al. (2021) for text-to-image synthesis. These methods share two common characteristics: (a) they employ autoregression over a *discrete* space, offering a probabilistic view of potential codes, and (b) they rely on *multiple* preceding values (up to $k^{th}$ order) through *complex* models, such as Transformer blocks.

In contrast, our work introduces a simple and first-order autoregressive approach, termed FINOLA (First-Order Norm+Linear Autoregression), designed to represent images. FINOLA operates by modeling images as feature maps and employing a straightforward autoregressive process. The key idea is that each pixel in the feature map depends solely on its immediate neighbor, simplifying the process to a first-order relationship. This is achieved by normalizing the values of the preceding neighbor and applying a shared linear model (norm+linear). Mathematically, it is represented as:

$$
\begin{aligned}
\boldsymbol{z}(x+1, y) &= \boldsymbol{z}(x, y) + \boldsymbol{A}\boldsymbol{z}_n(x, y) \\
\boldsymbol{z}(x, y+1) &= \boldsymbol{z}(x, y) + \boldsymbol{B}\boldsymbol{z}_n(x, y)
\end{aligned}
\quad where \quad
\boldsymbol{z}_n(x, y) = \frac{\boldsymbol{z}(x, y) - \mu_{\boldsymbol{z}}}{\sigma_{\boldsymbol{z}}},
\tag{1}
$$

where $\boldsymbol{z}$ represents the feature map with spatial dimensions $H \times W$ and $C$ channels ($\boldsymbol{z} \in \mathbb{R}^{H \times W \times C}$). The matrices $\boldsymbol{A}$ and $\boldsymbol{B}$ are both learnable, with dimensions $C \times C$. Notably, the feature $\boldsymbol{z}(x, y)$ is normalized across $C$ channels for each position $(x, y)$ by subtracting the mean $\mu_{\boldsymbol{z}}$ and dividing by the standard deviation $\sigma_{\boldsymbol{z}}$. An intriguing aspect is that the coefficient matrices $\boldsymbol{A}$ and $\boldsymbol{B}$ capture the relationship between each position and its rate of change, remaining invariant across different images. This underscores an intriguing intrinsic mathematical property shared by images within high-dimensional space. Moreover, the resulting feature map, obtained through autoregression, is *deterministically* used to predict the original image.

When extending Eq. 1 from a discrete image grid to continuous $x$ and $y$ coordinates, it unveils a mathematical description in the form of partial differential equations (PDEs):

$$
\frac{\partial \boldsymbol{z}}{\partial x} = \boldsymbol{A}\boldsymbol{z}_n, \quad \frac{\partial \boldsymbol{z}}{\partial y} = \boldsymbol{B}\boldsymbol{z}_n.
\tag{2}
$$

This mathematical representation offers a new insight, transcending individual instances. We validate FINOLA through two vision tasks: image reconstruction and self-supervised learning.

**Image reconstruction:** As illustrated in Figure 1, our approach begins by encoding the input image into a single vector $\boldsymbol{q}$ with $C$ channels. Subsequently, we generate the feature map $\boldsymbol{z} \in \mathbb{R}^{W \times H \times C}$ by placing $\boldsymbol{q}$ at the center of the feature map, i.e., $\boldsymbol{z}(\frac{W}{2}, \frac{H}{2}) = \boldsymbol{q}$, and then apply FINOLA to complete the feature map. Finally, we reconstruct the image to its original resolution using a decoder comprising upsampling and $3 \times 3$ convolutional layers. It's noteworthy that FINOLA can generate feature maps at various scales, ranging from $8 \times 8$ to $256 \times 256$ for an input image size of $256 \times 256$. In the most extreme case, where the feature map matches the resolution of the original image, the decoder only includes three $3 \times 3$ convolutional layers, highlighting FINOLA's effectiveness.

**Self-supervised pre-training:** FINOLA demonstrates potential for self-supervised pre-training. By employing a straightforward masked prediction task—in which a single unmasked quadrant block is encoded, and FINOLA is utilized to predict the masked region (as depicted in Figure 2)—our method achieves performance on par with established techniques (He et al. (2021); Xie et al. (2022)) in self-supervised learning. Additionally, masked FINOLA is well-suited for lightweight networks like Mobile-Former Chen et al. (2022), featuring 5.8M parameters and 285M FLOPs. It yields a highly

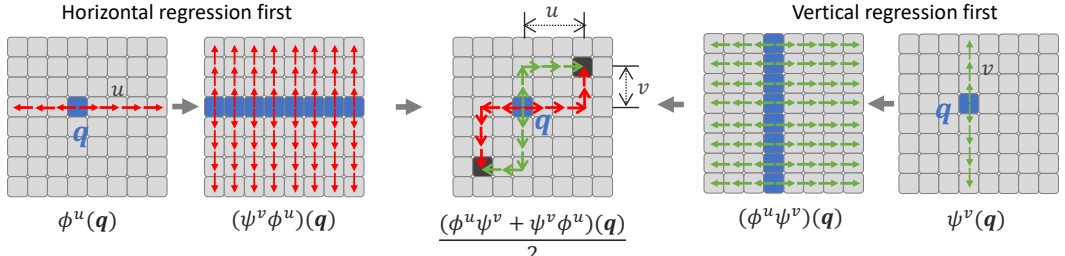

Figure 3: **Parallel implementation of FINOLA:** Horizontal and vertical regressions are separated. The *left* approach performs horizontal regression first, enabling parallel vertical regression. Similarly, the *right* approach starts with vertical regression, enabling parallel horizontal regression. The results of these approaches are averaged, corresponding to the two autoregression paths from the initial position marked by $q$. Best viewed in color.

capable encoder that can be shared for both image classification and object detection, eliminating the need for fine-tuning.

Comparing FINOLA and masked FINOLA, we observed that introducing masked prediction sacrifices restoration accuracy in favor of gaining semantic representation. This leads to notable improvements in both linear probing and fine-tuning for ImageNet classification. Additionally, masked FINOLA exhibits a significant increase in Gaussian curvature on the surfaces of critical features, indicating a greater curvature of the latent space for capturing semantics.

It's important to emphasize that our work does not aim to achieve state-of-the-art performance. Instead, our objective is to spotlight FINOLA as a mathematical property inherent in images, represented through partial differential equations (PDEs). We hope this contribution cultivate a deeper comprehension of images within the research community.

## 2 FINOLA: FIRST-ORDER NORM+LINEAR AUTOREGRESSION

This section offers a comprehensive explanation of the utilization of first-order norm+linear autoregression, known as FINOLA. Additionally, our approach unveils the extension of FINOLA into two partial differential equations (PDEs), providing insights into the underlying mathematical principles.

**FINOLA:** FINOLA generates a $W \times H$ feature map $z(x, y)$ autoregressively, starting from a single position, e.g., the center $z(\frac{W}{2}, \frac{H}{2})$. It is a ***first-order*** process, predicting each position using only its immediately previous neighbor. The prediction is ***simple***, achieved by applying a linear model on the normalized values of the previous neighbor.

This prediction is performed independently along the $x$ and $y$ axes using two separate linear models represented by $C \times C$ matrices, denoted as $A$ and $B$, as shown in Eq. 1. For predicting positions further away, such as $z(x + u, y)$ with $u > 1$, the process is repeated $u$ times as:

$$\begin{aligned} z(x + u, y) = \phi^u(z(x, y)) \\ z(x, y + v) = \psi^v(z(x, y)) \end{aligned} \quad where \ \ \phi(z) = z + \frac{A(z - \mu_z)}{\sigma_z}, \ \ \psi(z) = z + \frac{B(z - \mu_z)}{\sigma_z}, \quad (3)$$

where $\phi^u(\cdot)$ denotes applying $\phi$ by $u$ times, rather than indicating a power function. Importantly, the matrices $A$ and $B$, once learned from data, remain invariant across images, capturing the consistent relationship between the feature map's spatial derivatives and the feature values.

For diagonal predictions, e.g. from $z(x, y)$ to $z(x + u, y + v)$, the equations are combined as $(\phi^u \psi^v + \psi^v \phi^u)/2$. When predicting towards the left or up (with negative values of $u$ or $v$), we introduce two additional learnable matrices, $A_-$ and $B_-$, to perform predictions in the same manner as for right and down directions. Specifically, prediction toward the left is expressed as $z(x-1, y) = z(x, y) + A_- z_n(x, y)$.

**Parallel implementation:** Autoregression can be computationally intensive due to its sequential nature. FINOLA mitigates this by capitalizing on the independence of the $x$ and $y$ axes, enabling parallel execution, significantly boosting efficiency. As shown in Figure 3, performing horizontal regression first allows for parallel execution of subsequent vertical regression, and vice versa. In

practice, both approaches are combined by averaging their results. Each element in the result represents the average of the two autoregression paths originating from the initial position, marked as $\boldsymbol{q}$.

**Partial Differential Equations (PDEs):** Eq. 1 undergoes a transformation into a difference equation when we substitute $x + 1$ and $y + 1$ with $x + \Delta x$ and $y + \Delta y$, respectively, while letting $\Delta x$ and $\Delta y$ approach 1. As we further consider infinitesimal values for $\Delta x$ and $\Delta y$, we arrive at the formulation of partial differential equations (PDEs) as follows:

$$\begin{aligned}\frac{\boldsymbol{z}(x + \Delta x, y) - \boldsymbol{z}(x,y)}{\Delta x} &= \boldsymbol{A}\boldsymbol{z}_n(x,y) \\ \frac{\boldsymbol{z}(x, y + \Delta y) - \boldsymbol{z}(x,y)}{\Delta y} &= \boldsymbol{B}\boldsymbol{z}_n(x,y)\end{aligned} \qquad \xRightarrow{\Delta x \to 0,\ \Delta y \to 0} \qquad \begin{aligned}\frac{\partial \boldsymbol{z}}{\partial x} &= \boldsymbol{A}\boldsymbol{z}_n \\ \frac{\partial \boldsymbol{z}}{\partial y} &= \boldsymbol{B}\boldsymbol{z}_n\end{aligned}. \qquad (4)$$

These PDEs are inherently non-linear due to the normalization process. They represent a theoretical extension of FINOLA from a discrete grid to continuous coordinates. However, establishing their theoretical validity poses a substantial challenge. Instead, we present empirical evidence in subsequent experiments, demonstrating the effectiveness of FINOLA in generating feature maps for image reconstruction across a range of grid sizes, spanning from 8×8 to 256×256.

# 3    VALIDATION OF FINOLA

We validate FINOLA through two vision tasks: image reconstruction and self-supervised learning. In this section, we will delve into the application of FINOLA in these tasks.

## 3.1    FINOLA ON IMAGE RECONSTRUCTION

**Network architecture:** Utilizing FINOLA for image reconstruction is a straightforward process. As depicted in Figure 1, it begins by encoding the input image into a single vector $\boldsymbol{q}$ with $C$ channels. Next, we position $\boldsymbol{q}$ at the center of the feature map, i.e., $\boldsymbol{z}(\frac{W}{2}, \frac{H}{2}) = \boldsymbol{q}$, and apply FINOLA to complete the feature map $\boldsymbol{z} \in \mathbb{R}^{W \times H \times C}$. Finally, the original image is reconstructed by passing the feature map through a decoder, which consists of upsampling and 3×3 convolutional layers. Detailed architecture information for the encoder and decoder can be found in Appendix C.

**Multiple resolutions of generated feature maps:** FINOLA is capable of generating feature maps at various scales, ranging from 8×8 to 256×256 for an input image size of 256×256. It's important to note that as the resolution of feature maps increases, the corresponding decoder requires fewer upsample/convolutional blocks. In the most extreme case, when the feature map matches the resolution of the original image, the decoder includes only three 3×3 convolutional layers (see Table 8 in Appendix C). We intentionally reduce decoder complexity to evaluate FINOLA's performance in handling larger resolutions.

**Training Loss:** The entire network is trained end-to-end using mean square error over image pixels as the loss function.

## 3.2    MASKED FINOLA ON SELF-SUPERVISED PRE-TRAINING

FINOLA can be applied to self-supervised learning through a straightforward masked prediction task, which we refer to as *Masked FINOLA* to distinguish it from the vanilla FINOLA. Unlike vanilla FINOLA that support various resolutions of feature map, masked FINOLA performs mask prediction at resolution $\frac{1}{16}$, which is consistent with established baselines like MAE He et al. (2021), SimMIM Xie et al. (2022).

**Simple block masking:** FINOLA is applied through a simple masked prediction design that involves using a single unmasked image block (see Figure 4) to predict the surrounding masked region. Specifically, we crop out the unmasked block and pass it through the encoder, leveraging the power of FINOLA to generate a full-size feature map. Finally, a decoder is applied to recover the pixels in masked region. Unlike vanilla FINOLA, the reconstruction loss is computed only from the masked region. Please note that the unmasked block floats around the image randomly.

**Masked FINOLA variants:** Masked FINOLA comprises two variants: the element-wise approach (Masked-FINOLA-E) and the block-wise approach (Masked-FINOLA-B), as depicted in Figure 4.

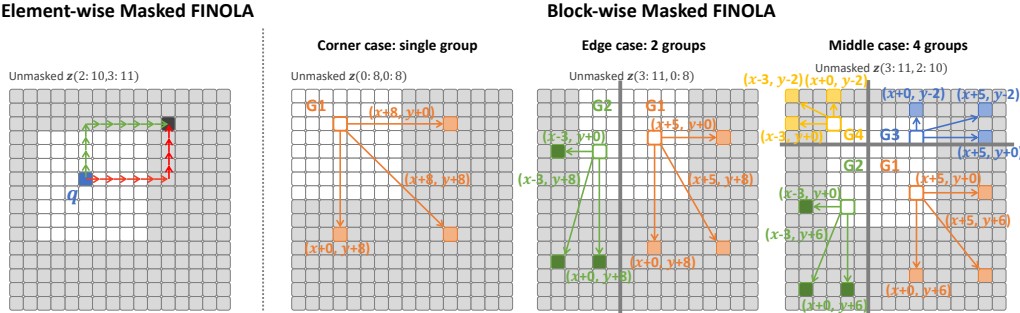

Figure 4: **Two Masked FINOLA variants:** element-wise (***left***) and block-wise (***right***) approaches. In the element-wise approach, autoregression is performed similarly to vanilla FINOLA, with the compressed vector $q$ observing only the unmasked block rather than the entire image. Conversely, the block-wise approach does not compress the unmasked block. Each unmasked position exclusively predicts three masked positions, as indicated by arrows, using Eq. 3. Assignments are grouped together, with shared offsets within each group. The grouping varies depending on the location of the unmasked quadrant, resulting in 1, 2, and 4 groups for corner, edge, and middle locations, respectively. Best viewed in color.

The element-wise variant (Masked-FINOLA-E) operates similarly to vanilla FINOLA, with the compressed vector $q$ only observing the unmasked block rather than the entire image (see Figure 4-left). To accommodate the longer training required in masked FINOLA (e.g., 1600 epochs), we follow He et al. (2021) to replace the convolutional decoder with a simple linear layer, transforming a $C$-channel token into a $16 \times 16 \times 3$ image patch.

In contrast, the block-wise variant (Masked-FINOLA-B) preserves the unmasked block in its entirety, without compression. It requires the unmasked block to have a quadrant size. As shown in Figure 4-right, each unmasked position is tasked with predicting three masked positions, denoted by arrows and computed using Eq. 3. These assignments are organized into groups, and within each group, all unmasked positions share common offsets for reaching their assigned masked positions. The configuration of these groups dynamically adapts based on the location of the unmasked quadrant, resulting in 1, 2, or 4 groups for corner, edge, or middle positions, respectively. To promote communication across these groups, transformer blocks are integrated into the decoder.

**Relation to MAE** He et al. (2021): Masked FINOLA shares a similar architecture with MAE but differs notably in ***masking*** and ***prediction*** strategies. Firstly, Masked FINOLA adopts a regular masking design, grouping all unmasked patches into a single block, in contrast to MAE's utilization of random unmasked patches. This design choice suits efficient CNN-based networks. Secondly, Masked FINOLA employs a first-order norm+linear autoregression approach for predicting the masked region, whereas MAE utilizes masked tokens within an attention model.

### 3.3 COMPARING FINOLA WITH MASKED FINOLA

In Table 1, we present a comparison between vanilla FINOLA and two masked FINOLA variants, assessing both their architectural distinctions and performance in image reconstruction and classification tasks. The introduction of masking, a characteristic of Masked FINOLA, entails a trade-off between restoration accuracy and enhanced semantic representation. Notably, among the masked FINOLA variants, the block-wise approach outperforms the element-wise counterpart, underscoring the challenges associated with masked prediction following compression. Figure 5 offers a geometric comparison. It reveals that Masked FINOLA exhibits a significant increase in Gaussian curvature on the surfaces of critical features, suggesting a greater curvature in the latent space for capturing semantics. Please see Appendix E for additional comparisons.

## 4 IMAGE RECONSTRUCTION EXPERIMENTS

We evaluate FINOLA for image reconstruction on ImageNet-1K Deng et al. (2009). For model and training details, please refer to Appendices C and D. Here, we summarize our key findings:

Table 1: **Comparing FINOLA and Masked FINOLA** on ImageNet-1K. Masked FINOLA variants trade restoration accuracy for enhanced semantic representation. The block-wise masked FINOLA outperforms the element-wise variant in linear probing (`lin`), probing with a single transformer block (`tran-1`), and fine-tuning (`tran-1-ft`).

| Model | Compress | Autoregression | Decoder | Recon-PSNR | lin | tran-1 | tran-1-ft |
|---|---|---|---|---|---|---|---|
| FINOLA | ✓ | element | up+conv | **25.8** | 17.9 | 46.8 | 81.9 |
| Masked FINOLA-E | ✓ | element | linear | 16.7 | 54.1 | 67.8 | 82.2 |
| Masked FINOLA-B | ✗ | block | trans+linear | 17.3 | **66.4** | **78.7** | **82.5** |

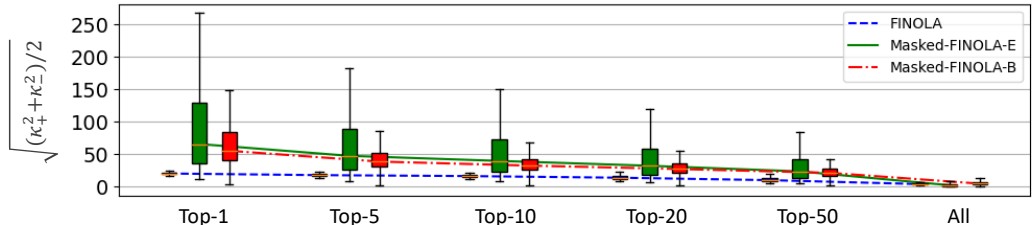

Figure 5: **Comparing Gaussian curvature of critical features: FINOLA vs. Masked FINOLA**. We evaluate this comparison using 50k IN-1K validation images, analyzing $16 \times 16$ surfaces in 3D space ($x$, $y$, $z_k$). Gaussian curvature is computed for all channels at each grid element. Channels within each image are sorted based on the root square mean of peak positive $\kappa_+$ and negative curvatures $\kappa_-$, and the distribution is plotted for the top-K features. Masked FINOLA demonstrates significantly larger curvature on critical features than vanilla FINOLA, highlighting the effectiveness of masked prediction in curving the latent space to capture semantics. Best viewed in color.

1. **Validation across multiple resolutions and image sizes:** FINOLA consistently performs well across various resolutions ($8 \times 8$ to $256 \times 256$) and different image sizes ($256 \times 256$ and $512 \times 512$).

2. **Validation of norm+linear model:** The combination of normalization and a linear model is crucial for success.

3. **Comparable performance with well-known baselines:** FINOLA achieves performance comparable to the first stage of VQGAN Esser et al. (2021) and stable diffusion Rombach et al. (2021) in image reconstruction. It also comparable with JPEG in image compression

4. **Comprehensive ablations and analysis:** Our ablation studies provide valuable insights into the control factors within FINOLA.

**Validation across multiple resolutions and image sizes:** We empirically validate the partial derivative equations (PDEs) in Eq. 2 by assessing FINOLA's performance in image reconstruction across various feature map resolutions and image sizes. Table 2-(a) displays reconstruction PSNR scores across different feature map resolutions. The reconstruction remains consistent across most resolutions, with slightly reduced performance at 128x128 and 256x256. This decrease is primarily due to significantly smaller decoders (with only 1.7M and 1.2M parameters, respectively). Notably, at 256x256 resolution, the feature map matches the image size, and the decoder comprises only three 3x3 convolutional layers to cover a 7-pixel field of view (see Table 8 in Appendix C).

Table 2-(b) presents results for three image sizes. FINOLA performs well on larger images, albeit with slightly lower PSNR scores, attributed to the higher compression rate of the encoder. In all cases, the encoder outputs a vector $q$ with 3072 channels, a dimension intentionally maintained to assess the model's ability to handle larger images with increased visual details.

**Validation of the norm+linear model:** To validate the *norm+linear* approach, we compared it with two alternative methods for completing the feature map: (a) repeating $q$ by $W \times H$ times and (b) a linear model without normalization. The reconstruction PSNR scores are reported in Table 3-(a). Repetition exhibits significantly lower scores, even though adding positional embedding provides a modest boost in performance. However, it still falls far behind norm+linear by a substantial margin, with a PSNR difference of 4.5 or more. The linear model alone experiences a slight performance drop at low resolution ($16 \times 16$), but it fails to converge at higher resolutions ($64 \times 64$). This clearly

Table 2: **Validation across multiple resolutions and image sizes:** PSNR values of image reconstruction are reported for multiple resolutions and image sizes on the ImageNet-1K validation set. The feature map has $C = 3072$ channels. Default resolution and image size are indicated with [†].

| Resolution | 8×8 | 16×16[†] | 32×32 | 64×64 | 128×128 | 256×256 | Image Size | 256×256[†] | 384×384 | 512×512 |
|---|---|---|---|---|---|---|---|---|---|---|
| Decoder | 25.3M | 18.5M | 9.6M | 7.9M | 1.7M | 1.2M | Feature Map | 16×16 | 24×24 | 32×32 |
| PSNR | 25.4 | 25.8 | 25.8 | 25.7 | 25.3 | 24.6 | PSNR | 25.8 | 24.5 | 24.2 |

(a) **Resolution of feature map $z$.**          (b) **Image size.**

Table 3: **Validation of norm+linear model:** PSNR values of image reconstruction are reported for alternative autogression and normalization models on the ImageNet-1K validation set. The feature map has $C = 3072$ channels. [‡] denotes the use of position embedding.

| Resolution | Repetition | Repetition[‡] | Linear | Norm+Linear | Stage | Batch-Norm | Layer-Norm |
|---|---|---|---|---|---|---|---|
| 16×16 | 16.1 | 20.2 | 25.4 | 25.8 | Training | 25.1 | 25.5 |
| 64×64 | 13.3 | 21.2 | not converge | 25.7 | Validation | 16.3 | 25.8 |

(a) **Autoregression models.**          (b) **Normalization models.**

demonstrates the critical importance of *norm+linear* for achieving successful reconstruction. Furthermore, Table 3-(b) highlights the significance of utilizing layer normalization. When replaced with batch normalization, the PSNR score during validation sees a significant drop, despite a less severe drop during training.

**Comparable performance with well-known baselines:** FINOLA not only uncovers mathematical invariances underlying images but also achieves performance on par with established baselines. Table 4-(a) presents a comparison of FINOLA with the first stage (autoencoding) of VQGAN Esser et al. (2021) and stable diffusion Rombach et al. (2021) in terms of image reconstruction. This evaluation is conducted on the ImageNet-Val dataset with a 256×256 image size. Remarkably, FINOLA attains higher PSNR scores while utilizing a smaller training dataset (1M images in ImageNet compared to 9M images in OpenImage) and a lower latent dimension. In Table 4-(b), we compare FINOLA with JPEG for image compression. Remarkably, by employing only uniform quantization per channel without further coding of the quantized bits, FINOLA achieves higher PSNR values with lower bits per pixel on both the ImageNet and Kodak Company (1999) datasets.

**Comprehensive ablations and analysis:** Our ablation studies and analysis shed light on key aspects of FINOLA. We summarize three ablations of hyper-parameters:

- The dimension of $q$ (or the number of channels) is critical, reaching a plateau when using more than 3072 channels (see Table 14-(a) and Figure 9 in Appendix F.1).

- The model size of the encoder is less critical but still related (see Table 14-(b) and Figure 10 in Appendix F.1).

- The position of placing $q$ is not critical (see Figure 11 in Appendix F.1).

We also made three intriguing observations about how images are distributed in the space of the compressed vector $q$ (referred to as the embedding space):

- The reconstruction from the averaged $\bar{q}$ over 50k images in the validation set results in a gray image (see Figure 13 in Appendix F.1).

- The space is predominantly occupied by noisy images (see Figure 12 in Appendix F.1).

- The reconstruction from an interpolation between two embeddings, $\alpha q_1 + (1 - \alpha) q_2$, yields a mix-up of corresponding images (see Figure 14 in Appendix F.1).

## 5 SELF-SUPERVISED PRE-TRAINING EXPERIMENTS

Our primary goal is to validate FINOLA as a fundamental mathematical property inherent in images. We aim to demonstrate its performance on par with established techniques like MAE He et al. (2021), SimMIM Xie et al. (2022), and MoCo Chen et al. (2020d). We focus on block-wise masked FINOLA (Masked-FINOLA-B) and evaluate its performance in ImageNet-1K classification and

Table 4: **Comparing FINOLA with baselines:** FINOLA's performance is compared to the first stage of VQGAN Esser et al. (2021) and stable diffusion Rombach et al. (2021) for image reconstruction on the ImageNet-1K validation set. Additionally, FINOLA's image compression results are compared to JPEG on ImageNet and Kodak Company (1999). It's worth noting that FINOLA employs uniform quantization per channel without further coding of the quantized bits.

| Model | Feature Dimension | Training Set | PSNR |
|---|---|---|---|
| VQGAN | 16×16×256 | OpenImage | 19.9 |
| Stable Diffusion | 16×16×256 | OpenImage | 24.1 |
| **FINOLA** | 1×1×3072 | ImageNet | 25.8 |

(a) **Image reconstruction evaluated on ImageNet-Val.**

| Method | ImageNet | | Kodak | |
|---|---|---|---|---|
| | Bit/Pixel | PSNR | Bit/Pixel | PSNR |
| JPEG | 0.50 | 24.5 | 0.20 | 24.0 |
| **FINOLA** | 0.19 | 24.9 | 0.19 | 25.6 |

(b) **Image compression.**

Table 5: **Comparison with masked encoding methods on ImageNet-1K using linear probing**. The baseline methods include iGPT Chen et al. (2020a), BEiT Bao et al. (2021), SimMIM Xie et al. (2022), MAE He et al. (2021) and MAE-Lite Wang et al. (2022). Three Mobile-Former backbones of varying widths are used. FINOLA pre-training demonstrates the ability to learn effective representations for small models. [†] denotes our implementation.

| Method | Model | Params | Top-1 |
|---|---|---|---|
| iGPT | iGPT-L | 1362M | 69.0 |
| BEiT | ViT-B | 86M | 56.7 |
| SimMIM | ViT-B | 86M | 56.7 |
| MAE | ViT-B | 86M | 68.0 |
| MAE[†] | ViT-S | 22M | 49.2 |
| MAE-Lite | ViT-Tiny | 6M | 23.3 |
| **FINOLA** | MF-W720 | 6M | 51.3 |
| **FINOLA** | MF-W1440 | 14M | 62.8 |
| **FINOLA** | MF-W2880 | 28M | 66.4 |

Table 6: **Comparison with previous self-supervised methods on ImageNet-1K fine-tuning**. The baseline methods includes MoCo-v3 Chen* et al. (2021), MAE-Lite Wang et al. (2022), UM-MAE Li et al. (2022b), MAE He et al. (2021), and SimMIM Xie et al. (2022). Three Mobile-Former backbones of varying widths are used, followed by a `tran-4` decoder with 4 transformer blocks.

| Method | Model | MAdds | Params | Top-1 |
|---|---|---|---|---|
| MoCo-v3 | ViT-Tiny | 1.2G | **6M** | 76.8 |
| MAE-Lite | ViT-Tiny | 1.2G | **6M** | 78.0 |
| **FINOLA** | MF-W720 | **0.7G** | 7M | **78.4** |
| MoCo-v3 | ViT-S | 4.6G | 22M | 81.4 |
| UM-MAE | Swin-T | 4.5G | 29M | 82.0 |
| MAE-Lite | ViT-S | 4.6G | 22M | 82.1 |
| SimMIM | Swin-T | 4.5G | 29M | **82.2** |
| **FINOLA** | MF-W1440 | **2.6G** | **20M** | 82.2 |
| MoCo-v3 | ViT-B | 16.8G | 86M | 83.2 |
| MAE | ViT-B | 16.8G | 86M | 83.6 |
| SimMIM | ViT-B | 16.8G | 86M | 83.8 |
| SimMIM | Swin-B | 15.4G | 88M | **84.0** |
| **FINOLA** | MF-W2880 | **9.9G** | **57M** | 83.9 |

COCO object detection. For brevity, we refer to Masked FINOLA-B as "FINOLA" throughout this section. Please refer to Appendices C, D, and G for network structure, training setup, and additional experiments, respectively. Here are our key findings:

**FINOLA achieves comparable performance with established baselines (e.g., MAE, SimMIM):** We evaluate FINOLA as complete end-to-end systems and compare them with baselines. For example, we compare FINOLA+MobileFormer with MAE+ViT in the context of ImageNet classification. Our comparisons include linear probing (Table 5) and fine-tuning (Table 6). FINOLA achieves comparable performance on both evaluations while requiring lower FLOPs.

**FINOLA provides a robust task-agnostic encoders:** Pre-training with FINOLA followed by fine-tuning on ImageNet-1K (IN-1K) consistently outperforms IN-1K supervised pre-training in both ImageNet classification and COCO object detection (see Figure 6). The gains in object detection are substantial, ranging from 5 to 6.4 AP. Remarkably, even without IN-1K fine-tuning, FINOLA pre-training alone outperforms the supervised counterpart in object detection by a clear margin (3 to 4.5 AP, detailed in the Appendix). This highlights FINOLA's ability to encode spatial structures.

When compared to MoCo-V2 (as shown in Table 17 in Appendix G.3), FINOLA demonstrates comparable performance in linear probing while surpassing MoCo-V2 in IN-1K fine-tuning, COCO object detection, and instance segmentation. FINOLA's superior performance suggests its effective encoding of spatial structures, supporting the idea that the underlying PDEs capture intrinsic spatial structures present in images.

**FINOLA has strong performance when fine-tuning on COCO:** see details in Appendix G.4.

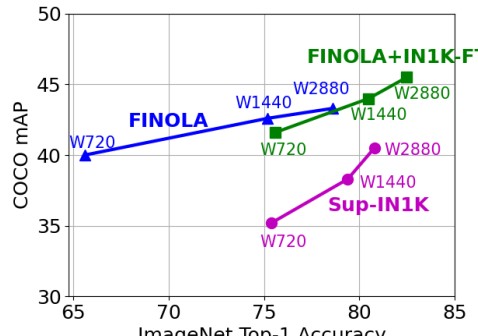

Figure 6: **Task-agnostic encoders** evaluated on ImageNet (IN-1K) classification and COCO object detection. We assess three IN-1K pretraining methods: (a) supervised (Sup-IN1K), (b) FINOLA, and (c) FINOLA with fine-tuning on IN-1K (FINOLA+IN1K-FT). The dots represent different Mobile-Former backbones. For classification, we add a `tran-1` decoder (with a single transformer block) trained with class supervision. It's important to note that the backbone remains task-agnostic, frozen during object detection. FINOLA performs lower than Sup-IN1K in classification but surpasses it in object detection. After fine-tuning on IN-1K, FINOLA+IN1K-FT shows improvements in both tasks, providing robust task-agnostic encoders.

## 6 RELATED WORK

**Image autoregression** van den Oord et al. (2016b;a); Salimans et al. (2017); Chen et al. (2018); Yu et al. (2022b) use conditional probability distributions to generate high-quality images based on previously generated pixels. These models have evolved from pixel-level focus to operating in the latent space using vector quantization van den Oord et al. (2017); Razavi et al. (2019); Esser et al. (2021). In contrast, we present a first-order norm+linear autoregression to generate feature map.

**Masked image modeling** (MIM) is inspired by the success of BERT Devlin et al. (2019) and ViT Dosovitskiy et al. (2021) to learn representation by predicting masked region from unmasked counterpart. BEiT Bao et al. (2021) and PeCo Dong et al. (2021) predict on tokens, MaskFeat Wei et al. (2022) predicts on HOG, and MAE He et al. (2021) reconstructs original pixels. Recent works further explore combining MIM and contrastive learning Zhou et al. (2022); Dong et al. (2022); Huang et al. (2022); Tao et al. (2022); Assran et al. (2022); Jiang et al. or techniques suitable for ConvNets Gao et al. (2022); Jing et al. (2022); Fang et al. (2022). Different from these works that use random masking, FINOLA uses regular masking and simpler norm+linear prediction.

**Contrastive methods:** Becker & Hinton (1992); Hadsell et al. (2006); van den Oord et al. (2018); Wu et al. (2018); He et al. (2019); Chen & He (2020); Caron et al. (2021) achieve significant progress. They are most applied to Siamese architectures Chen et al. (2020b); He et al. (2019); Chen et al. (2020d); Chen* et al. (2021) to contrast image similarity and dissimilarity and rely on data augmentation. Chen & He (2020); Grill et al. (2020) remove dissimilarity between negative samples by handling collapse carefully. Chen et al. (2020c); Li et al. (2021a) show pre-trained models work well for semi-supervised learning and few-shot transfer.

## 7 CONCLUSION

This paper introduces FINOLA, a novel framework that represents every image as a first-order norm+linear autoregressive process. This discovery unveils the presence of underlying partial differential equations (PDEs) governing the latent feature space. We validate FINOLA through experiments in image reconstruction and self-supervised learning, demonstrating its remarkable capability to autoregress feature maps up to the original image's resolution. This empowers successful image reconstruction using a minimalist decoder architecture. Additionally, we leverage FINOLA for self-supervised learning by employing a straightforward masked prediction approach. Our findings reveal that this pre-trained representation excels in various downstream tasks, including image classification and object detection, without the need for extensive fine-tuning. In summary, FINOLA provides valuable mathematical insights into the realm of image representations.

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

## A    CALCULATION OF GAUSSIAN CURVATURE

To compute the Gaussian curvature, we consider the feature map per channel as a set of $W \times H$ surfaces $z_k(x, y)$ in 3D space, where $x$, $y$, and $z_k$ denote the coordinates. At each position $(x, y)$, the Gaussian curvature for the $k^{th}$ channel can be determined using the following equation:

$$\kappa_k(x, y) = \frac{\frac{\partial^2 z_k}{\partial x^2} \frac{\partial^2 z_k}{\partial y^2} - \left(\frac{\partial^2 z_k}{\partial x \partial y}\right)^2}{\left(1 + (\frac{\partial z_k}{\partial x})^2 + (\frac{\partial z_k}{\partial y})^2\right)^2}. \tag{5}$$

Subsequently, we rank the channels based on the root mean square of the peak positive curvature ($\kappa_+$) and the peak negative curvature ($\kappa_-$) over the surface.

## B    LIMITATIONS

The major limitation of our image reconstruction method is the loss of high-frequency details, as demonstrated in Figure 1. The resulting images exhibit blurred faces, trees, and deformed small texts. This limitation may be attributed to the choice of loss function, as we currently use the mean square error. In future work, we plan to explore the use of adversarial loss, as suggested in VQGAN Esser et al. (2021), to promote high-quality reconstruction and address this limitation.

## C    NETWORK ARCHITECTURES

In this section, we provide detailed information on the network architecture components used in our study. Specifically, we describe (a) the Mobile-Former encoders, (b) the pooler to compress the feature map into a single vector, (c) the decoders employed in both FINOLA and Masked FINOLA, (d) the decoders designed for image classification, and (e) the decoders tailored for object detection.

**Mobile-Former encoders:** Mobile-Former Chen et al. (2022) is used as the encoder in our approach. It is a CNN-based network that extends MobileNet Sandler et al. (2018) by adding 6 global tokens in parallel. To preserve spatial details, we increase the resolution of the last stage from $\frac{1}{32}$ to $\frac{1}{16}$. We evaluate three variants of Mobile-Former, which are detailed in Table 7. Each variant consists of 12 blocks and 6 global tokens, but they differ in width (720, 1440, 2880). These models serve as the encoders (or backbones) for image reconstruction, self-supervised pre-training, and

Table 7: **Specification of Mobile-Former encoders**. "bneck-lite" denotes the lite bottleneck block Li et al. (2021b). "M-F" denotes the Mobile-Former block and "M-F$^\downarrow$" denotes the Mobile-Former block for downsampling.

| Stage | Resolution | Block | MF-W2880 | | MF-W1440 | | MF-W720 | |
|---|---|---|---|---|---|---|---|---|
| | | | #exp | #out | #exp | #out | #exp | #out |
| token | | | 6×256 | | 6×256 | | 6×192 | |
| stem | $256^2$ | conv 3×3 | – | 64 | – | 32 | – | 16 |
| 1 | $128^2$ | bneck-lite | 128 | 64 | 64 | 32 | 32 | 16 |
| 2 | $64^2$ | M-F$^\downarrow$ | 384 | 112 | 192 | 56 | 96 | 28 |
| | | M-F | 336 | 112 | 168 | 56 | 84 | 28 |
| 3 | $32^2$ | M-F$^\downarrow$ | 672 | 192 | 336 | 96 | 168 | 48 |
| | | M-F | 576 | 192 | 288 | 96 | 144 | 48 |
| | | M-F | 576 | 192 | 288 | 96 | 144 | 48 |
| 4 | $16^2$ | M-F$^\downarrow$ | 1152 | 352 | 288 | 96 | 240 | 80 |
| | | M-F | 1408 | 352 | 704 | 176 | 320 | 88 |
| | | M-F | 1408 | 352 | 704 | 176 | 480 | 88 |
| | | M-F | 2112 | 480 | 1056 | 240 | 528 | 120 |
| | | M-F | 2880 | 480 | 1440 | 240 | 720 | 120 |
| | | M-F | 2880 | 480 | 1440 | 240 | 720 | 120 |
| | | conv 1×1 | – | 2880 | – | 1440 | – | 720 |

Table 8: **Decoder specifications**. The decoder's complexity decreases as the spatial resolution increases from 8×8 to 256×256). "res-conv" represents a residual block He et al. (2016) consisting of two 3x3 convolutional layers, while "up-conv" performs upsampling followed by a 3x3 convolutional layer.

| Resolution | 8×8 block | #out | 16×16 block | #out | 32×32 block | #out | 64×64 block | #out | 128×128 block | #out | 256×256 block | #out |
|---|---|---|---|---|---|---|---|---|---|---|---|---|
| $8^2$ | res-conv | 512 | | | | | | | | | | |
| $16^2$ | up-conv | 512 | | | | | | | | | | |
| | res-conv | 512 | res-conv | 512 | | | | | | | | |
| $32^2$ | up-conv | 512 | up-conv | 512 | | | | | | | | |
| | res-conv | 256 | res-conv | 256 | res-conv | 256 | | | | | | |
| $64^2$ | up-conv | 256 | up-conv | 256 | up-conv | 256 | | | | | | |
| | res-conv | 256 | res-conv | 256 | res-conv | 256 | res-conv | 256 | | | | |
| $128^2$ | up-conv | 256 | up-conv | 256 | up-conv | 256 | up-conv | 256 | | | | |
| | res-conv | 128 | res-conv | 128 | res-conv | 128 | res-conv | 128 | res-conv | 128 | | |
| $256^2$ | up-conv | 128 | up-conv | 128 | up-conv | 128 | up-conv | 128 | up-conv | 128 | | |
| | res-conv | 128 | res-conv | 128 | res-conv | 128 | res-conv | 128 | res-conv | 128 | res-conv | 128 |
| | conv3×3 | 3 | conv3×3 | 3 | conv3×3 | 3 | conv3×3 | 3 | conv3×3 | 3 | conv3×3 | 3 |
| #param | 25.3M | | 18.5M | | 9.6M | | 7.9M | | 1.7M | | 1.2M | |

Table 9: **Mobile-Former decoder specifications for COCO object detection:** 100 object queries with dimension 256 are used. "down-conv" includes a 3×3 depthwise convolution (stride=2) and a pointwise convolution (256 channels). "up-conv" uses bilinear interpolation, followed by a 3×3 depthwise and a pointwise convolution. "M-F$^+$" replaces the *Mobile* sub-block with a transformer block, while "M-F$^-$" uses the lite bottleneck Li et al. (2021b) to replace the *Mobile* sub-block.

| Stage | MF-Dec-522 | | MF-Dec-211 | |
|---|---|---|---|---|
| query | 100×256 | | 100×256 | |
| $\frac{1}{32}$ | down-conv | | down-conv | |
| | M-F$^+$ | ×5 | M-F$^+$ | ×2 |
| $\frac{1}{16}$ | up-conv | | up-conv | |
| | M-F$^-$ | ×2 | M-F$^-$ | ×1 |
| $\frac{1}{8}$ | up-conv | | up-conv | |
| | M-F$^-$ | ×2 | M-F$^-$ | ×1 |

evaluation in image classification and object detection tasks. For image reconstruction, we also explore two wider models, W4320 and W5760, which increase the number of channels from W2880 by 1.5 and 2 times, respectively. It's important to note that these models were manually designed without an architectural search for optimal parameters such as width or depth.

**Pooling the compressed vector $q$:** In both FINOLA and element-wise masked FINOLA, the compressed vector $q$ is obtained by performing attentional pooling Lee et al. (2019); Yu et al. (2022a) on the feature map. This pooling operation involves a single multi-head attention layer with learnable queries, where the encoder output serves as both the keys and values.

**Decoders for FINOLA pre-training:** Table 8 provides the architecture details of the decoders used in FINOLA. The complexity of the decoder decreases as the spatial resolution increases, going from 8×8 to 256×256. Unlike vanilla FINOLA, which employs stacked upsampling and convolution blocks, the Masked FINOLA variants utilize simpler architectures—a linear layer for transforming features into 16×16 image patches. This choice facilitates longer training. As mentioned in the main paper, the decoder of Masked-FINOLA-B incorporates transformer blocks (without positional embedding) to enable spatial communication. It's worth noting that vanilla FINOLA is trained for 100 epochs on ImageNet, while Masked FINOLA undergoes training for 1600 epochs.

**Decoders for ImageNet classification:** We utilize three decoders to evaluate the pre-trained encoders in FINOLA. These decoders are as follows:

- `lin` decoder: It consists of a single linear layer and is used for linear probing.

Table 10: **Pre-training setting for FINOLA and masked FINOLA variants.**

| Config | FINOLA | Masked FINOLA |
|---|---|---|
| optimizer | AdamW | AdamW |
| base learning rate | 1.5e-4 | 1.5e-4 |
| weight decay | 0.1 | 0.1 |
| batch size | 128 | 1024 |
| learning rate schedule | cosine decay | cosine decay |
| warmup epochs | 10 | 10 |
| training epochs | 100 | 1600 |
| image size | $256^2$ | $256^2$ |
| augmentation | RandomResizeCrop | RandomResizeCrop |

Table 11: **Settings for linear probing and `tran-1` probing on ImageNet-1K:** The encoders are frozen during both tasks.

| Config | Linear probing | `tran-1` probing |
|---|---|---|
| optimizer | SGD | AdamW |
| base learning rate | 0.1 | 0.0005 |
| weight decay | 0 | 0.1 |
| batch size | 4096 | 4096 |
| learning rate schedule | cosine decay | cosine decay |
| warmup epochs | 10 | 10 |
| training epochs | 90 | 200 |
| augmentation | RandomResizeCrop | RandAug (9, 0.5) |
| label smoothing | – | 0.1 |
| dropout | – | 0.1 (MF-W720) 0.2 (MF-W1440/W2880) |
| random erase | – | 0 (MF-W720/W1440) 0.25 (MF-W2880) |

- `tran-1` decoder: It incorporates a shallower transformer decoder with a single transformer block followed by a linear classifier and is employed for `tran-1` probing and fine-tuning.
- `tran-4` decoder: This decoder is composed of four transformer blocks followed by a linear classifier and is utilized for fine-tuning alone.

The transformer decoders are designed with different widths (192, 384, 768) to correspond with the three Mobile-Former encoders, which have widths of 720, 1440, and 2880, respectively.

**Decoders for object detection:** The decoders used in the DETR framework with Mobile-Former Chen et al. (2022) are described in Table 9. Both decoders consist of 100 object queries with a dimension of 256. While they share a similar structure across three scales, they differ in terms of their depths. Since the backbone network ends at a resolution of $\frac{1}{16}$, the decoder incorporates a downsampling step to further reduce the resolution to $\frac{1}{32}$. This enables the decoder to efficiently process the features for object detection.

## D  TRAINING SETUP

In this section, we provide detailed training setups for different tasks, including:

- Image reconstruction using FINOLA on ImageNet-1K.
- Masked FINOLA pre-training on ImageNet-1K.
- Linear probing on ImageNet-1K.
- `tran-1` probing on ImageNet-1K.
- Fine-tuning on ImageNet-1K.
- COCO object detection.

**FINOLA pre-training:** The pre-training settings for image compression and reconstruction using FINOLA are provided in Table 10. The learning rate is scaled as $lr = base\_lr \times$ batchsize / 256.

Table 12: **Setting for end-to-end fine-tuning on ImageNet-1K.**

| Config | Value |
|---|---|
| optimizer | AdamW |
| base learning rate | 0.0005 |
| weight decay | 0.05 |
| layer-wise lr decay | 0.90 (MF-W720/W1440) 0.85 (MF-W2880) |
| batch size | 512 |
| learning rate schedule | cosine decay |
| warmup epochs | 5 |
| training epochs | 200 (MF-W720) 150 (MF-W1440) 100 (MF-W2880) |
| augmentation | RandAug (9, 0.5) |
| label smoothing | 0.1 |
| mixup | 0 (MF-W720) 0.2 (MF-W1440) 0.8 (MF-W2880) |
| cutmix | 0 (MF-W720) 0.25 (MF-W1440) 1.0 (MF-W2880) |
| dropout | 0.2 |
| random erase | 0.25 |

**Masked FINOLA pre-training:** Similar to the vanilla FINOLA, masked FINOLA also follows the training setup described in Table 10, but with a larger batch size due to the simpler decoder architecture that requires less memory consumption.

**Linear probing:** In our linear probing, we follow the approach described in He et al. (2021) by incorporating an additional BatchNorm layer without affine transformation (affine=False). Detailed settings can be found in Table 11.

**`tran-1` probing:** The settings for `tran-1` decoder probing are presented in Table 11. It is important to note that the default decoder widths are 192, 384, and 768 for MF-W720, MF-W1440, and MF-W2880, respectively.

**End-to-end fine-tuning on ImageNet-1K:** The settings for the end-to-end fine-tuning of both the encoder and `tran-1` decoder are presented in Table 12. The decoder weights are initialized from the `tran-1` probing stage.

**Decoder probing on COCO object detection:** In this configuration, the backbone pre-trained on ImageNet-1K is frozen, and only the decoders are trained for 500 epochs on 8 GPUs with 2 images per GPU. We employ AdamW optimizer with an initial learning rate of 1e-4. The learning rate is decreased by a factor of 10 after 400 epochs. The weight decay is 1e-4, and the dropout rate is 0.1.

**Fine-tuning on COCO object detection:** In this setting, both the encoder and decoder are fine-tuned. The fine-tuning process consists of an additional 200 epochs following the decoder probing stage. The initial learning rate for both the encoder and decoder is set to 1e-5, which decreases to 1e-6 after 150 epochs.

# E  ADDITIONAL COMPARISON BETWEEN FINOLA AND MASKED FINOLA

**Comparison of FINOLA and masked FINOLA on ImageNet classification:** Table 13 presents the results of linear and `tran-1` probing applied to the vanilla FINOLA across various dimensions of the latent space. Notably, even the highest accuracy achieved by the vanilla FINOLA falls significantly behind both masked FINOLA variants (element-wise or block-wise). This stark difference highlights the remarkable power of masked prediction in learning semantic representations.

**Comparison of FINOLA and masked FINOLA on image reconstruction:** Figure 7 presents a comparison of reconstructed samples obtained using FINOLA and masked FINOLA. In the case of the two masked FINOLA variants (element-wise and block-wise), the encoders are frozen, and only their attentional pooling and FINOLA components are fine-tuned. To ensure a fair comparison, we utilize the same architecture for the decoders in the masked FINOLA variants as in FINOLA, training them from scratch. The corresponding peak signal-to-noise ratio (PSNR) values on the ImageNet validation set are provided at the bottom. While the masked variants preserve color and shape information, they exhibit a loss of texture details compared to the vanilla FINOLA. Notably, as demonstrated in the main paper, the masked FINOLA variants demonstrate stronger semantic

Table 13: **FINOLA vs. masked FINOLA on ImageNet Deng et al. (2009) classification:** Compared to masked FINOLA variants, FINOLA performs poorly on both linear probing (`lin`) and probing with a single transformer block (`tran-1`) with clear margins. Even we search over the dimension of latent space from 64 to 3072, the gap is still large, i.e. more than 20%. Block-wise masked FINOLA (Masked-FINOLA-B) outperforms the element-wise variant (Masked-FINOLA-E), achieving higher accuracy. Please note that the encoders are frozen when performing linear and `tran-1` probing.

| Pre-training | Dim of $q$ | lin | tran-1 |
|---|---|---|---|
| FINOLA | 64 | 10.2 | 20.2 |
| | 128 | 11.5 | 24.0 |
| | 256 | 15.0 | 29.0 |
| | 512 | 20.1 | 34.1 |
| | 1024 | 23.0 | 39.6 |
| | 2048 | 23.2 | 41.1 |
| | 3072 | 17.9 | 46.8 |
| Masked FINOLA-E | 512 | 54.1 | 67.8 |
| Masked FINOLA-B | — | 66.4 | 78.7 |

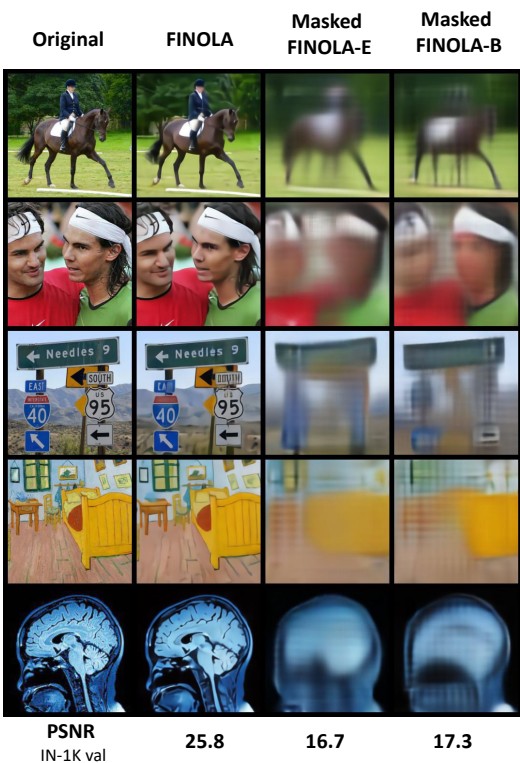

Figure 7: **FINOLA vs. masked FINOLA on image reconstruction:** In this comparison, the encoders of the two masked FINOLA variants are frozen, and their attentional pooling and FINOLA components are fine-tuned. To ensure a fair comparison, we replace the decoders in the masked FINOLA variants with the same architecture as FINOLA, trained from scratch. When compared to vanilla FINOLA, the masked variants preserve color and shape information but exhibit a loss of texture details.

Table 14: **Image reconstruction ablation experiments** on ImageNet-1K. We report PSNR on the validate set. The reconstruction quality correlates to (a) the number of channels in the latent space and (b) complexity of encoder. Default settings are marked by [†].

| #Channels | 4096 | 3072[†] | 2048 | 1024 | 512 | 256 | 128 | 64 |
|---|---|---|---|---|---|---|---|---|
| PSNR | 25.9 | 25.8 | 25.1 | 23.7 | 22.2 | 20.8 | 19.4 | 18.2 |

(a) **Number of channels in latent space.**

| Encoder | 67.6M | 43.5M | 25.0M[†] | 12.0M | 5.0M |
|---|---|---|---|---|---|
| PSNR | 26.1 | 26.0 | 25.8 | 25.1 | 24.4 |

(b) **Model size of encoders.**

representation. This comparison highlights that FINOLA and masked FINOLA adhere to the same mathematical principles (involving partial differential equations) but strike different balances between semantic representation and preserving fine details.

**Comparison between two Masked FINOLA variants:** Figure 8 showcases the results of linear probing, `tran-1` probing, and fine-tuning for two masked FINOLA variants trained with different schedules. The block-wise masked FINOLA consistently outperforms its element-wise counterpart across all evaluations. These findings demonstrate the effectiveness of directly applying FINOLA on the unmasked features to predict the masked region, as opposed to performing compression before applying FINOLA.

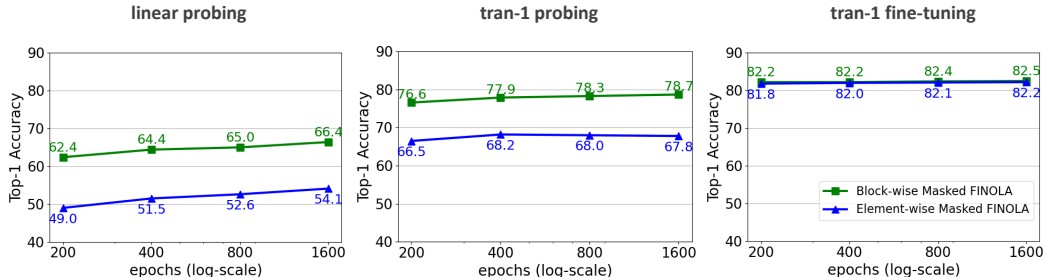

Figure 8: **Comparison of element-wise and block-wise masked FINOLA**. The evaluation includes linear probing, `tran-1` probing, and `tran-1` fine-tuning. Block-wise masked FINOLA consistently outperforms the element-wise counterpart across all evaluations. Notably, the performance gap in fine-tuning is smaller compared to linear and `tran-1` probing. Best viewed in color.

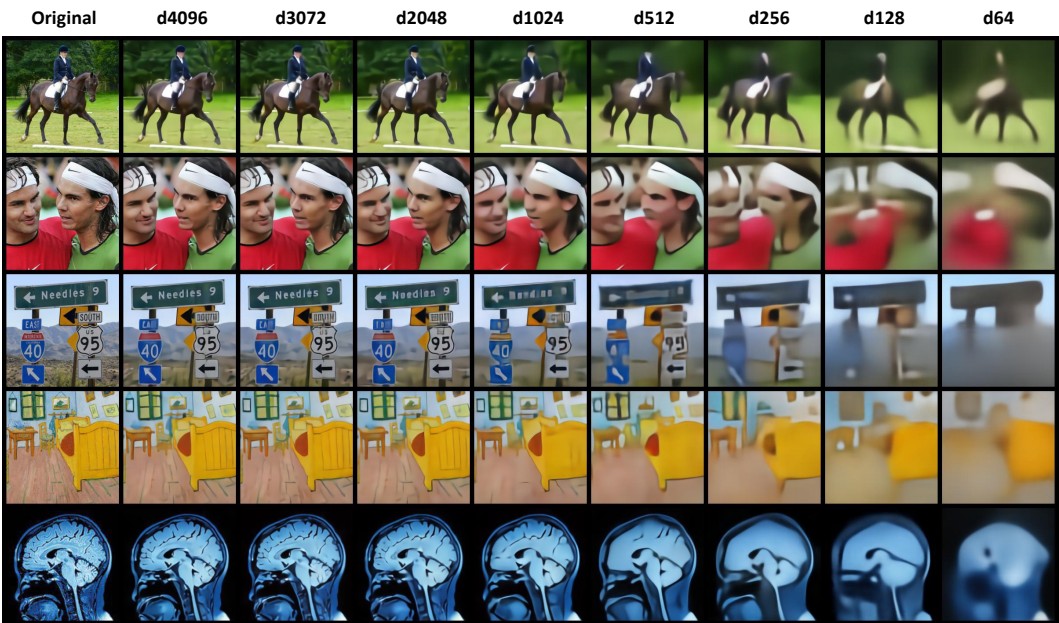

Figure 9: **Image reconstruction examples.** The leftmost column shows the original images. The number of channels in the latent space, decreasing from 4096 to 64 from the left to right, controls the reconstruction quality. Best viewed in color.

## F  ADDITIONAL EXPERIMENTAL RESULTS ON IMAGE RECONSTRUCTION

In this section, we present additional experimental results on image reconstruction.

### F.1  ABLATION STUDIES

**The number of channels in the latent space is crucial.** Table 14-(a) presents the PSNR values for various latent space dimensions, while Figure 9 showcases the corresponding reconstructed examples. The image quality is noticeably poor when using only 64 channels, resulting in significant loss of details. However, as the number of channels increases, more details are successfully recovered. Using more than 3072 channels yields reasonably good image quality, achieving a PSNR of 25.8.

**The model size of encoder is less critical but also related.** As shown in Figure 10 and Table 14-(b), the larger model has better image quality. But the gap is not significant. When increasing model size by 13 times from 5.0M to 67.6M, the PSNR is slightly improved from 24.4 to 26.1. Note all encoders share similar architecture (Mobile-Former with 12 blocks), but have different widths.

**The position of $q$ is not critical:** Figure 11 showcases the reconstructed samples obtained by placing the compressed vector $q$ at different positions, including the center and four corners. The

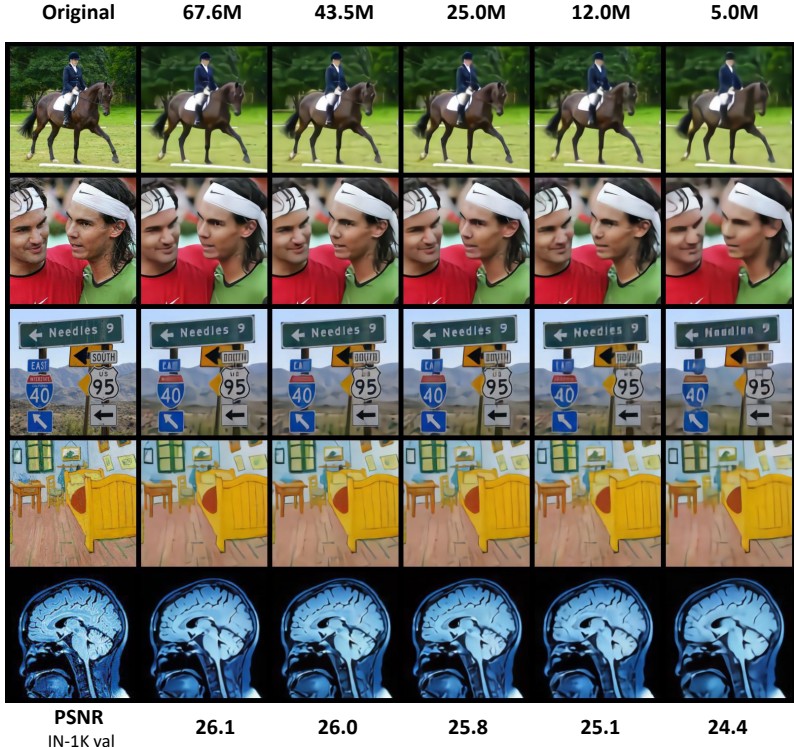

Figure 10: **Impact of encoder size on image reconstruction quality:** The image reconstruction quality shows a slight improvement as the size of the encoder increases. Even with a small encoder containing 5 million parameters (right column), it effectively compresses an image into a single vector capable of reconstructing the entire image. Best viewed in color.

corresponding peak signal-to-noise ratio (PSNR) values on the ImageNet validation set are provided at the bottom. While placing $q$ at the center yields slightly better results compared to corner positions, the difference is negligible. It is important to note that each positioning corresponds to its own pre-trained model with non-shared parameters.

### F.2 INSPECTING THE EMBEDDING SPACE

In this subsection, we list main observations and analysis in the space of the compressed vector $q$ (named embedding space). This will help us to understand how images are distributed in the embedding space.

**Three observations:** Below we list three observations that reveal properties of the embedding space.

*Dominance of noisy images in the space:* To analyze the distribution of images in the embedding space, we collected $q$ vector for all 50,000 images from the ImageNet validation set and computed their statistics (mean and covariance). By sampling embeddings based on these statistics and reconstructing images, we consistently observed the emergence of similar noisy patterns, as depicted in Figure 12. This observation highlights the prevalence of noisy images throughout the space, with good images appearing as isolated instances surrounded by the abundance of noise.

*Averaged embedding $\bar{q}$ yields a gray image:* In Figure 13, we observe that the reconstructed image obtained from the averaged embedding $\bar{q}$, computed over 50,000 images from the ImageNet validation set, closely resembles a gray image. We further investigate the relationship between real image embeddings $q$ and the averaged embedding $\bar{q}$ through interpolations along the embedding space. As depicted in the ***left*** figure, the reconstructed images maintain their content while gradually fading into a gray image. Additionally, we extend this connection to mirror embeddings in the ***right*** figure, represented by $2q - \bar{q}$, which correspond to images with reversed colors. These findings suggest

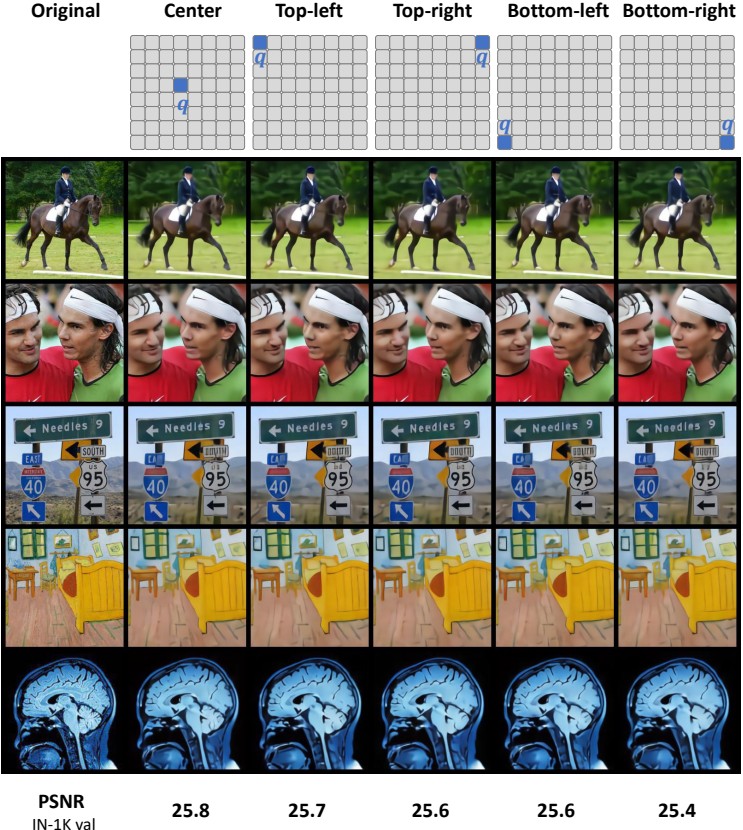

Figure 11: **Comparison of different positions of compressed vector $q$:** The quality of image reconstruction shows minimal sensitivity to the position of $q$. Placing it at the center yields slightly better results compared to corner positions. It is worth noting that each positioning has its own pre-trained model with non-shared parameters. Best viewed in color.

that despite the prevalence of noisy images, the line segment connecting an image embedding to the average embedding encompasses different color transformations of the same image.

*Reconstruction from interpolated embeddings:* In Figure 14, we present the reconstructed images obtained by interpolating between two image embeddings using the equation $\alpha q_1 + (1 - \alpha) q_2$. This process of embedding mixup results in a corresponding mixup of the images, allowing for a smooth transition between the two original images by varying the value of $\alpha$. However, it is important to note that the resulting reconstruction may not precisely match the simple mixup of the original images, represented by $\alpha I_1 + (1 - \alpha) I_2$.

Combining the three observations discussed above, our findings suggest that the presence of noisy images in Figure 12 indicates the mixing of multiple surrounding images. As the number of image embeddings involved in the mixing process increases, the resulting reconstructions tend to resemble a gray image, as depicted in Figure 13.

**Principle component analysis (PCA):** The reconstruction results shown in Figure 15 are obtained using PCA with the top-$K$ principle components. These components correspond to the largest $K$ eigenvalues of the covariance matrix computed from 50,000 image embeddings in the ImageNet validation set. The principle components capture essential information, starting with color and layout, and gradually encoding finer image details as more components are included in the reconstruction process.

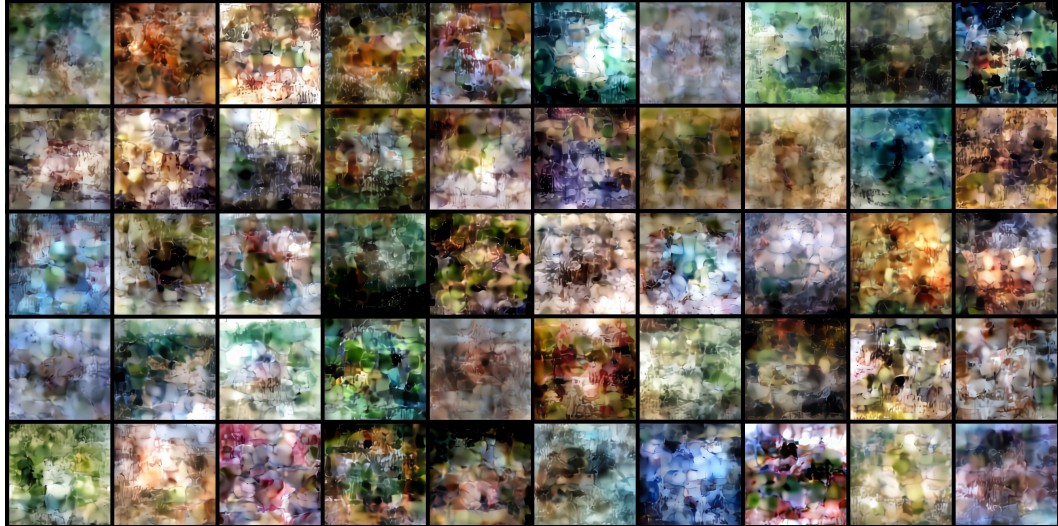

Figure 12: **Reconstruction from random samples:** The reconstructed images are generated by sampling from the statistics (mean and covariance) of compressed embeddings $q$ obtained from the ImageNet validation set, consisting of 50,000 images. Although the samples are not similar to images of Gaussian noise, they lack semantic meaning and appear as noisy images. Multiple samplings consistently yield similar noisy patterns. Best viewed in color.

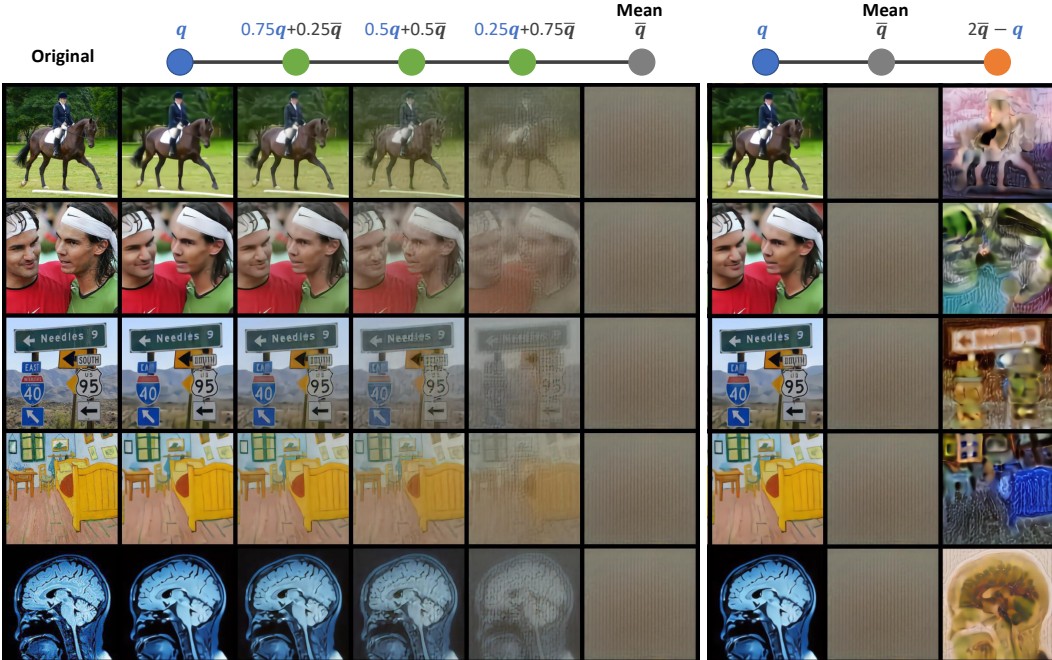

Figure 13: **Reconstruction from the average embedding $\bar{q}$:** The reconstructed image corresponding to the average embedding $\bar{q}$ computed from 50,000 ImageNet validation images closely resembles a gray image (shown in the right column of the left figure). In the *left* figure, we demonstrate the interpolation along a line connecting embeddings from different images to the average embedding. Notably, the reconstructed images progressively fade into a gray image. In the *right* figure, we extend the connection between an image embedding $q$ and the average embedding $\bar{q}$ to a mirror embedding $2\bar{q} - q$, corresponding to an image with reversed colors. This comparison provides insights into the nature of the embedding space. Best viewed in color.

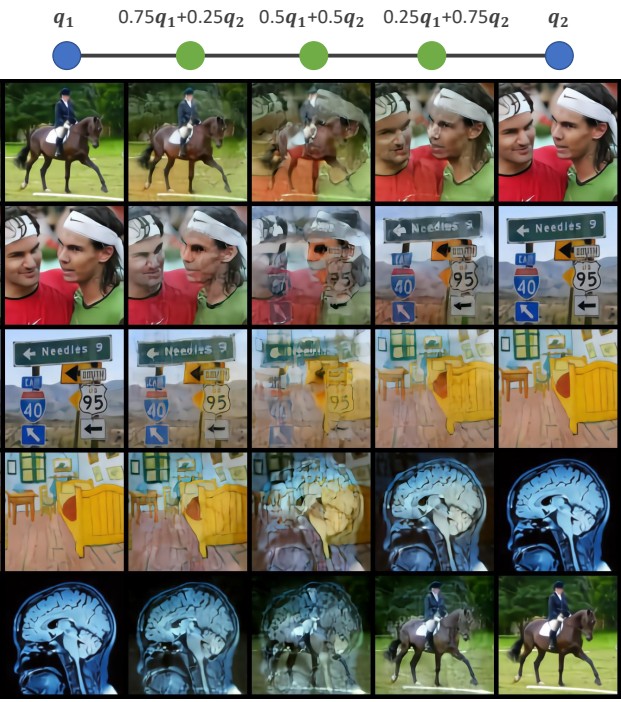

Figure 14: **Reconstruction from interpolated embeddings:** The images are reconstructed by interpolating embeddings of two images, $\alpha \boldsymbol{q}_1 + (1 - \alpha)\boldsymbol{q}_2$. Although the mixed embedding passes through a non-linear network that includes FINOLA and a multi-layer decoder, it leads to mixing up images as output. Best viewed in color.

## G ADDITIONAL EXPERIMENTS ON SELF-SUPERVISED PRE-TRAINING

In this section, we present more ablations on block-wise masked FINOLA (Masked-FINOLA-B) and additional comparisons between Masked-FINOLA-B and baselines. For brevity, we will use the term "FINOLA" to refer to Masked-FINOLA-B throughout the remainder of this section.

### G.1 ABLATION STUDIES

**Ablation on training schedule:** The impact of training schedule length on three Mobile-Former encoders is depicted in Figure 16. Notably, the accuracies of both linear and `tran-1` probings demonstrate a consistent improvement as the training duration increases. Interestingly, even with a pre-training of just 100 epochs, fine-tuning with `tran-1` achieves commendable performance. This finding diverges from the observations in MAE He et al. (2021), where longer training is essential for fine-tuning improvements.

**Ablation on the number of transformer blocks in the decoder:** We investigate the impact of the number of transformer blocks in the decoder on FINOLA pre-training using the Mobile-Former-W2880 as encoder. Each transformer block in the decoder consists of 512 channels, but does *not* use positional embedding. The results, shown in Table 15, demonstrate that adding more transformer blocks leads to consistent improvements in both linear and `tran-1` probing tasks. However, we observe that the performance of fine-tuning is less sensitive to changes in the decoder depth.

### G.2 COMPARABLE PERFORMANCE WITH ESTABLISHED BASELINES

**Comparison with additional baselines on ImageNet fine-tuning:** The fine-tuning results of FI-NOLA are compared with those of previous self-supervised methods in Table 16. For this comparison, FINOLA utilizes the Mobile-Former-W1440 encoder, followed by a `tran-4` decoder consisting of four transformer blocks with 384 channels. The results demonstrate that FINOLA achieves comparable performance to the baselines while requiring lower floating-point operations (FLOPs).

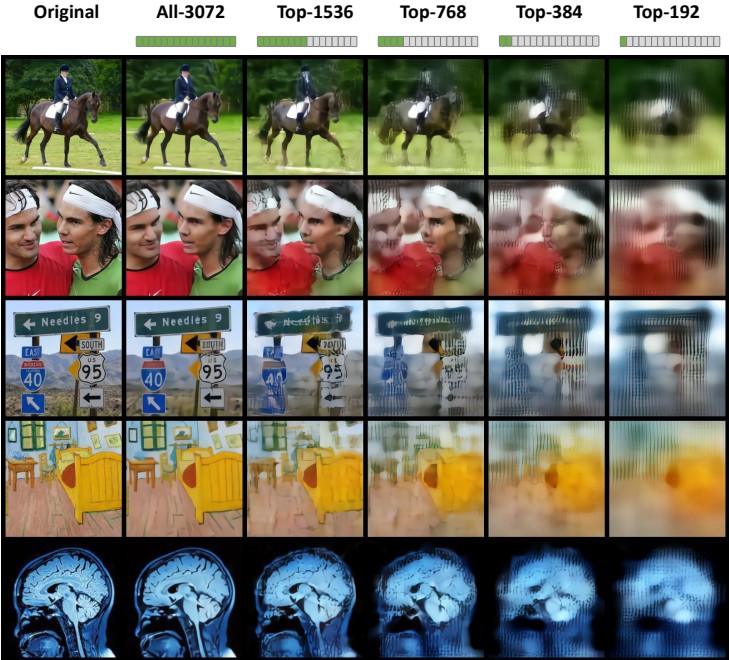

Figure 15: **Reconstruction from top principle components:** The top-$K$ principle components correspond to the largest $K$ eigenvalues of the covariance matrix computed from 50,000 image embeddings in the ImageNet validation set. With a selection of top-192 components (the right column), the color and layout of the images are primarily determined, but the resulting reconstructions appear blurred with noticeable loss of details. As more principle components are incorporated, the finer details are gradually restored. Best viewed in color.

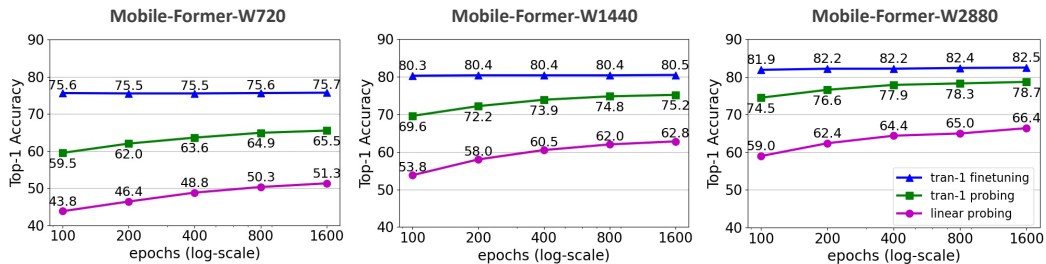

Figure 16: **Training schedules of Masked-FINOLA-B.** Longer training schedule provides consistent improvement for linear and `tran-1` probing over different models, while fine-tuning performance is not sensitive to training schedule. Best viewed in color.

This highlights the effectiveness and efficiency of FINOLA in the context of self-supervised pre-training.

### G.3 ROBUST TASK AGNOSTIC ENCODERS

**Comparisons with MoCo-v2**: As shown in Table 17, FINOLA demonstrates comparable performance to MoCo-V2 in linear probing, while surpassing MoCo-V2 in `tran-1` probing that uses a single transformer block as a decoder for classification, IN-1K fine-tuning, object detection and segmentation. The backbone is frozen for both COCO object detection and segmentation. FINOLA's superior performance suggests it learns more effective intermediate features, contributing to more representative decoder features. Furthermore, the improved performance in object detection emphasizes FINOLA's ability to encode spatial structures effectively.

Table 15: **Ablation on the number of transformer blocks in the decoder:** Evaluation is conducted on ImageNet using Mobile-Former-W2880 as the encoder. Each transformer block consists of 512 channels. Each model is pre-trained for 800 epochs. Increasing the decoder depth exhibits consistent improvement for linear and `tran-1` probing, while fine-tuning performance shows limited sensitivity to decoder depth.

| #Blocks | lin | tran-1 | tran-1-ft |
|---------|------|--------|-----------|
| 1 | 61.1 | 74.4 | 82.2 |
| 2 | 62.6 | 76.5 | 82.3 |
| 3 | 63.5 | 77.3 | 82.2 |
| 4 | 63.8 | 78.0 | 82.3 |
| 5 | 64.0 | 78.1 | 82.3 |
| 6 | 65.0 | 78.3 | 82.4 |

Table 16: **Comparison with self-supervised baselines on ImageNet-1K fine-tuning.** The baseline methods includes iBOT Zhou et al. (2022), MoCo-v3 Chen* et al. (2021), MAE He et al. (2021), MAE-Lite Wang et al. (2022), CMAE Huang et al. (2022), and ConvMAE Gao et al. (2022). Mobile-Former-W1440 (pre-trained for 1600 epochs) is used as encoder, followed by a `tran-4` decoder with 4 transformer blocks with 384 channels.

| Method | Model | MAdds | Params | Top-1 |
|--------|-------|-------|--------|-------|
| iBOT | ViT-S | 4.6G | 22M | 82.3 |
| MoCo-v3 | ViT-S | 4.6G | 22M | 81.4 |
| MAE | ViT-S | 4.6G | 22M | 79.5 |
| MAE-Lite | ViT-S | 4.6G | 22M | 82.1 |
| CMAE | ViT-S | 4.6G | 22M | 80.2 |
| ConvMAE | ConvViT-S | 6.4G | 22M | **82.6** |
| **FINOLA** | MF-W1440 | **2.6G** | **20M** | 82.2 |

Table 17: **Comparisons with MoCo-v2 Chen et al. (2020d)** on ImageNet classification, COCO object detection and instance segmentation. Three Mobile-Former backbones with different widths are used. In `tran-1`, the encoder is frozen while a transformer block is trained as a decoder using class labels. In `tran-1-ft`, encoders are fine-tuned. Encoders are frozen in both COCO object detection and instance segmentation. DETR framework is used for object detection, while Mask-RCNN ($1\times$) is used for segmentation. FINOLA outperforms MoCo-V2 in most evaluations, except on par in linear probing.

| Pre-training | Encoder | IN-1K Top-1 | | | COCO Det (Box-AP) | | COCO Seg (Mask-AP) | |
|--------------|---------|------|--------|-----------|-----------|-----------|-----------|-----------|
| | | lin | tran-1 | tran-1-ft | w/o IN-ft | with IN-ft | w/o IN-ft | with IN-ft |
| MoCo-V2 | MF-W720 | **51.6** | 52.9 | 74.3 | 31.8 | 39.9 | 23.2 | 25.3 |
| **FINOLA** | | 51.3 | **65.5** | **75.6** | **40.0** | **41.6** | **26.3** | **28.4** |
| MoCo-V2 | MF-W1440 | 60.4 | 58.5 | 79.2 | 30.3 | 39.0 | 25.6 | 25.7 |
| **FINOLA** | | **62.8** | **75.2** | **80.5** | **42.6** | **44.0** | **30.6** | **32.7** |
| MoCo-V2 | MF-W2880 | **66.5** | 63.8 | 80.0 | 25.5 | 31.7 | 27.8 | 25.2 |
| **FINOLA** | | 66.4 | **78.7** | **82.5** | **43.3** | **45.5** | **33.3** | **35.1** |

These experiments demonstrate that the proposed masked FINOLA is able to learn task-agnostic representation by using a simple masking design. This supports that the underling PDEs capture the intrinsic spatial structures present in images.

**Comparison with the IN-1K supervised pre-training on transferring to COCO object detection:** Table 18 presents the results of COCO object detection using frozen backbones. The evaluation utilizes three Mobile-Former encoders with different widths and two Mobile-Former decoders with different depths. Notably, FINOLA pre-training followed by ImageNet-1K (IN-1K) fine-tuning consistently outperforms the IN-1K supervised pre-training across all evaluations, demonstrating the effectiveness of task-agnostic encoders. Impressively, even FINOLA pre-training alone, without IN-1K fine-tuning, surpasses the supervised counterpart on object detection by a significant margin of 2.6–5.2 AP. This showcases FINOLA's ability to encode spatial structures.

## G.4 FINE-TUNING ON COCO

Furthermore, fine-tuning the backbone on COCO further enhances detection performance. Table 19 provides a comprehensive comparison of fine-tuning results using the Mobile-Former Chen et al. (2022) in the DETR Carion et al. (2020) framework. Unlike the frozen backbone configuration, where FINOLA outperforms supervised pre-training significantly (as shown in Table 18), they achieve similar performance in COCO fine-tuning. This is because the advantage of FINOLA pre-training on spatial representation diminishes when object labels in COCO provide strong guidance. However, FINOLA maintains its leading position by leveraging fine-tuning on IN-1K to improve semantic representation and transfer it to object detection. Compared to the supervised baseline,

Table 18: **COCO object detection results** on the `val2017` dataset using a ***frozen*** backbone pre-trained on ImageNet-1K. Evaluation is conducted over three backbones and two heads that use Mobile-Former Chen et al. (2022) end-to-end in DETR Carion et al. (2020) framework. Our FI-NOLA consistently outperform the supervised counterpart. Notably, fine-tuning on ImageNet-1K (denoted as "IN-ft") yields further improvements. The initial "MF" (e.g., `MF-Dec-522`) denotes Mobile-Former. The madds metric is based on an image size of 800×1333.

| model | Head madds (G) | param (M) | model | Backbone madds (G) | param (M) | pre-train | IN-ft | AP | AP$_{50}$ | AP$_{75}$ | AP$_S$ | AP$_M$ | AP$_L$ |
|---|---|---|---|---|---|---|---|---|---|---|---|---|---|
| MF Dec 522 | 34.6 | 19.4 | MF W2880 | 77.5 | 25.0 | supervised | – | 40.5 | 58.5 | 43.3 | 21.1 | 43.4 | 56.8 |
| | | | | | | FINOLA | ✗ | 43.3 (+2.8) | 61.5 | 46.8 | 23.7 | 46.9 | 60.1 |
| | | | | | | FINOLA | ✓ | **45.5** (+5.0) | **63.8** | **49.5** | **25.1** | **49.1** | **63.5** |
| | 32.3 | 18.6 | MF W1440 | 20.4 | 11.7 | supervised | – | 38.3 | 56.0 | 40.8 | 19.0 | 40.9 | 54.3 |
| | | | | | | FINOLA | ✗ | 42.6 (+4.3) | 60.3 | 46.1 | 22.6 | 46.2 | 60.0 |
| | | | | | | FINOLA | ✓ | **44.0** (+5.7) | **62.3** | **47.3** | **23.8** | **47.6** | **61.0** |
| | 31.1 | 18.2 | MF W720 | 5.6 | 4.9 | supervised | – | 35.2 | 52.1 | 37.6 | 16.9 | 37.2 | 51.7 |
| | | | | | | FINOLA | ✗ | 40.0 (+4.8) | 57.9 | 42.9 | 20.6 | 43.3 | 56.8 |
| | | | | | | FINOLA | ✓ | **41.6** (+6.4) | **59.4** | **45.0** | **21.2** | **45.0** | **58.9** |
| MF Dec 211 | 15.7 | 9.2 | MF W2880 | 77.5 | 25.0 | supervised | – | 34.1 | 51.3 | 36.1 | 15.5 | 36.8 | 50.0 |
| | | | | | | FINOLA | ✗ | 36.7 (+2.6) | 53.7 | 39.3 | 18.2 | 39.7 | 52.2 |
| | | | | | | FINOLA | ✓ | **41.0** (+6.9) | **59.2** | **44.4** | **20.9** | **44.6** | **58.3** |
| | 13.4 | 8.4 | MF W1440 | 20.4 | 11.7 | supervised | – | 31.2 | 47.8 | 32.8 | 13.7 | 32.9 | 46.9 |
| | | | | | | FINOLA | ✗ | 36.0 (+4.8) | 52.7 | 38.7 | 16.6 | 39.1 | 52.5 |
| | | | | | | FINOLA | ✓ | **39.2** (+8.0) | **56.9** | **42.0** | **19.7** | **42.8** | **56.2** |
| | 12.2 | 8.0 | MF W720 | 5.6 | 4.9 | supervised | – | 27.8 | 43.4 | 28.9 | 11.3 | 29.1 | 41.6 |
| | | | | | | FINOLA | ✗ | 33.0 (+5.2) | 49.3 | 35.0 | 15.3 | 35.1 | 48.9 |
| | | | | | | FINOLA | ✓ | **35.8** (+8.0) | **52.6** | **38.3** | **16.4** | **38.3** | **52.0** |

Table 19: **COCO object detection results** on the `val2017` dataset after ***fine-tuning*** both the backbone and head on COCO. Evaluation is performed on three different backbones and two heads, utilizing the Mobile-Former Chen et al. (2022) end-to-end in the DETR Carion et al. (2020) framework. Our approach, which involves FINOLA pre-training followed by ImageNet-1K fine-tuning, surpasses the performance of the supervised baselines. The initial "MF" (e.g., `MF-Dec-522`) denotes Mobile-Former, while "IN-ft" indicates fine-tuning on ImageNet-1K. The reported madds values are based on the image size of 800×1333.

| model | Head madds (G) | param (M) | model | Backbone madds (G) | param (M) | pre-train | IN-ft | AP | AP$_{50}$ | AP$_{75}$ | AP$_S$ | AP$_M$ | AP$_L$ |
|---|---|---|---|---|---|---|---|---|---|---|---|---|---|
| MF Dec 522 | 34.6 | 19.4 | MF W2880 | 77.5 | 25.0 | supervised | – | 48.1 | 66.6 | 52.5 | 29.7 | 51.8 | 64.0 |
| | | | | | | FINOLA | ✗ | 48.0 (-0.1) | 66.2 | 52.3 | 28.2 | 51.4 | 64.1 |
| | | | | | | FINOLA | ✓ | **49.0** (+0.9) | **67.7** | **53.4** | **30.1** | **52.9** | **65.5** |
| | 32.3 | 18.6 | MF W1440 | 20.4 | 11.7 | supervised | – | 46.2 | 64.4 | 50.1 | 27.1 | 49.8 | 62.4 |
| | | | | | | FINOLA | ✗ | 46.8 (+0.6) | 64.9 | 51.0 | 26.6 | 50.6 | 63.4 |
| | | | | | | FINOLA | ✓ | **47.3** (+1.1) | **65.6** | **51.4** | **27.3** | **50.7** | **63.9** |
| | 31.1 | 18.2 | MF W720 | 5.6 | 4.9 | supervised | – | 42.5 | 60.4 | 46.0 | 23.9 | 46.0 | 58.5 |
| | | | | | | FINOLA | ✗ | 43.3 (+0.8) | 61.0 | 47.0 | 23.1 | 46.6 | 61.0 |
| | | | | | | FINOLA | ✓ | **44.4** (+1.9) | **62.1** | **48.1** | **24.3** | **47.8** | **61.5** |
| MF Dec 211 | 15.7 | 9.2 | MF W2880 | 77.5 | 25.0 | supervised | – | 44.0 | 62.8 | 47.7 | 25.8 | 47.3 | 60.7 |
| | | | | | | FINOLA | ✗ | 44.4 (+0.4) | 62.5 | 48.2 | 24.7 | 47.6 | 60.7 |
| | | | | | | FINOLA | ✓ | **46.0** (+2.0) | **64.8** | **49.9** | **26.2** | **50.0** | **62.7** |
| | 13.4 | 8.4 | MF W1440 | 20.4 | 11.7 | supervised | – | 42.5 | 60.6 | 46.0 | 23.6 | 45.9 | 57.9 |
| | | | | | | FINOLA | ✗ | 42.4 (-0.1) | 60.2 | 45.9 | 21.9 | 45.7 | 60.0 |
| | | | | | | FINOLA | ✓ | **43.8** (+1.3) | **61.8** | **47.5** | **23.9** | **47.1** | **60.8** |
| | 12.2 | 8.0 | MF W720 | 5.6 | 4.9 | supervised | – | 37.6 | 55.1 | 40.4 | 18.9 | 40.6 | 53.8 |
| | | | | | | FINOLA | ✗ | 37.2 (-0.4) | 54.3 | 39.7 | 18.7 | 39.8 | 53.4 |
| | | | | | | FINOLA | ✓ | **39.3** (+1.7) | **56.7** | **42.4** | **19.4** | **42.1** | **56.5** |

FINOLA pre-training followed by IN-1K fine-tuning achieves a gain of 0.9–2.0 AP for all three encoders and two decoders.

Table 20: **Comparison with DETR-based models on COCO detection.** All baselines are fine-tuned on COCO. FINOLA-DETR utilizes Mobile-Former (MF-W2880) as the backbone, which has similar FLOPs and model size to the ResNet-50 used in other methods. MAdds are calculated based on an image size of 800×1333.

| Model | Query | AP | $AP_{50}$ | $AP_{75}$ | $AP_S$ | $AP_M$ | $AP_L$ | MAdds (G) | Param (M) |
|---|---|---|---|---|---|---|---|---|---|
| DETR-DC5Carion et al. (2020) | **100** | 43.3 | 63.1 | 45.9 | 22.5 | 47.3 | 61.1 | 187 | 41 |
| Deform-DETRZhu et al. (2020) | 300 | 46.2 | 65.2 | 50.0 | 28.8 | 49.2 | 61.7 | 173 | 40 |
| DAB-DETRLiu et al. (2022) | 900 | 46.9 | 66.0 | 50.8 | 30.1 | 50.4 | 62.5 | 195 | 48 |
| DN-DETRLi et al. (2022a) | 900 | 48.6 | 67.4 | 52.7 | 31.0 | 52.0 | 63.7 | 195 | 48 |
| DINOZhang et al. (2022) | 900 | **50.9** | **69.0** | **55.3** | **34.6** | **54.1** | 64.6 | 279 | 47 |
| **FINOLA-DETR (frozen)** | **100** | 45.5 | 63.8 | 49.5 | 25.1 | 49.1 | 63.5 | **112** | 44 |
| **FINOLA-DETR (fine-tune)** | | 49.0 | 67.7 | 53.4 | 30.1 | 52.9 | **65.5** | | |

Table 20 compares FINOLA-DETR (in which the backbone is fine-tuned in the DETR framework) with existed DETR baselines. FINOLA-DETR achieves an AP of 49.0, outperforming most DETR-based detectors except DINO Zhang et al. (2022). Remarkably, our method achieves these results while using significantly fewer FLOPs (112G vs. 279G) and object queries (100 vs. 900). When compared to DETR-DC5 with a fine-tuned backbone, FINOLA-DETR with a *frozen* backbone achieves a 2.2 AP improvement while reducing MAdds by 40%.

These results showcase the efficacy of FINOLA in capturing rich image representations even with more compact models, offering a promising approach for efficient self-supervised learning.

