# OpenReview forum: "Image as First-Order Norm+Linear Autoregression: Unveiling Mathematical Invariance"
_ICLR.cc/2024/Conference — Submitted to ICLR 2024_

### Official Review · Reviewer_QCbo · 2023-10-31

**Soundness:** 2 fair
**Presentation:** 2 fair
**Contribution:** 2 fair
**Rating:** 3
**Confidence:** 4

**Summary:**

This paper presents a First-Order Norm+Linear Autoregression method (called FINOLA) which represents each image in the latent space as a first-order autoregressive process. Then, the authors validate the FINOLA property from on image reconstruction and self-supervised learning. Experiments demonstrate the effectiveness of the proposed method.

**Strengths:**

This paper represents each image in the latent space as a first-order autoregressive process, and conducts experiments on image reconstruction and self-supervised learning to validate the method.

**Weaknesses:**

Some details of the figures and the method are not clear. The experiment section can be improved.

**Questions:**

1. Some details of the figures are not clear. For example, what do red and blue arrows in z(x, y) mean? How do they affect the reconstruction? Does a single vector q mean the feature of the whole image or the specific pixel? In the figure, the a high frequency in the reconstruction loss.

2. How to calculate the mean $\mu_z$ and the standard deviation $\sigma_z$? In Eqn. (1), how to initialize the matrix A and B? What are the constraints of the matrix A and B?

3. The authors mainly compare VQGAN (Esser et al. (2021)) and stable diffusion (Rombach et al. (2021)) in image reconstruction. It would be better to compare recent image reconstruction methods. In addition, could you compare convolutional U-Net for image reconstruction?

4. In Table 4 (a), using PSNR to compare VQGAN and Stable Diffusion is not convincing because these methods are generative methods. It would be better to use LPIPS and FID.

---

> ### Author Response · Authors · 2023-11-17
> **Authors' Response (part 1)**
>
> Thank you for dedicating your time and effort to provide feedback on our work. Below, we answer the questions that have been raised.
>
> ---
> **Q1: Some details of the figures are not clear.**
>
> **For example, what do red and blue arrows in z(x, y) mean?**
> The red and blue arrows in $\mathbfit{z}(x, y)$ represent the two autoregression paths of FINOLA for any given position $(x, y)$ from the center where the global pooled vector $\mathbfit{q}$ is placed. Each arrow signifies a FINOLA step, and the color denotes the order of regression. The red path involves horizontal regression followed by vertical regression, while the green path involves vertical regression followed by horizontal regression. The results of these two paths are averaged to obtain the final feature vector.
>
> **How do they affect the reconstruction?**
> It's crucial to note that we repeat this "two-path" autoregression until the entire feature map is generated. Subsequently, the feature map goes through the decoder to reconstruct the images.
>
> **Does a single vector q mean the feature of the whole image or the specific pixel?**
> *"A single vector q"* indeed denotes the feature of the *entire image* after pooling.
>
> **In the figure, the a high frequency in the reconstruction loss.**
> As for the concern about the loss of high-frequency details in the reconstructed images, we acknowledge this limitation, and it is discussed in detail in Appendix B. We attribute this limitation to the choice of the $L_2$ loss function. Our intentional use of a simple loss function is to highlight that FINOLA is not heavily dependent on complex reconstruction loss functions. To improve image quality, an alternative approach could involve exploring the use of a combination of perceptual loss and an adversarial training procedure with a patch-based discriminator, as outlined in the VQGAN framework.
>
> ---
> **Q2: How to calculate the mean $\mu_z$ and the standard deviation $\sigma_z$? In Eqn. (1), how to initialize the matrix A and B? What are the constraints of the matrix A and B?**
>
> The mean $\mu_z$ and the standard deviation $\sigma_z$ are calculated per position $(x, y)$ on the feature map $\mathbfit{z}$. For each position $(x, y)$, the corresponding feature $\mathbfit{z}(x, y)$ is a vector with $C$ channels: $\mathbfit{z}(x, y)=[z_1,\dots,z_C]^T$. $\mu_z$ and $\sigma_z$ are the mean $\mu_z=\frac{1}{C}\sum z_i$ and standard deviation $\sigma_z=\sqrt{\frac{\sum (z_i-\mu_z)^2}{C}}$ of these $C$ values.
>
> Matrices $\mathbfit{A}$ and $\mathbfit{B}$ are randomly initialized, and no specific constraints are placed on them.

---

> ### Author Response · Authors · 2023-11-17
> **Authors' Response (part 2)**
>
> **Q3: The authors mainly compare VQGAN (Esser et al. (2021)) and stable diffusion (Rombach et al. (2021)) in image reconstruction. It would be better to compare recent image reconstruction methods. In addition, could you compare convolutional U-Net for image reconstruction?**
>
> ***Comparison with convolutional U-Net:*** The table below provides a comparative analysis between our method and convolutional U-Net for image reconstruction (measured by PSNR). Both approaches share the same Mobile-Former encoder and have identical bottleneck dimensions (C=1024 or 3072). In our method, FINOLA is employed to generate a 16x16 feature map, followed by a convolutional decoder to reconstruct an image with a size of 256x256. On the other hand, U-Net has a deeper decoder that utilizes convolution and upsampling for image reconstruction. Both methods boast a similar model size. The results showcase the superior performance of FINOLA over convolutional U-Net, indicating that a single layer FINOLA is more effective than multi-layer convolution and upsampling.
>
> |Method|C=1024|C=3072|
> |---|---|---|
> | Convolutional U-Net | 23.4 | 25.1 |
> | **FINOLA (our)** | **23.7** | **25.8** |
>
> ***Comparison with ViT-VQGAN:*** In the following comparison, Masked FINOLA is evaluated against ViT-VQGAN (*"Vector-Quantized Image Modeling with Improved VQGAN, Yu et al., ICLR 2022"*) using ImageNet linear probing. FINOLA achieves higher accuracy with a much more compact encoder.
>
> |Method|Params|Top-1|
> |---|---|---|
> | ViT-VQGAN | 650M |  65.1 |
> | **FINOLA (our)** | **28M** | **66.4** |
>
> For a detailed comparison in image reconstruction, we will address this in the subsequent question (**Q4**).
>
> ---
> **Q4: In Table 4 (a), using PSNR to compare VQGAN and Stable Diffusion is not convincing because these methods are generative methods. It would be better to use LPIPS and FID.**
>
> Below, we present a comparison of FINOLA with the ***first stage*** of multiple generative methods, assessed using both PSNR and FID metrics. While FINOLA outperforms in PSNR, it lags in FID. This discrepancy can be attributed to our deliberate use of the $L_2$ loss function. We intentionally opted for a simple loss function to underscore that FINOLA does not heavily rely on intricate reconstruction loss functions. To enhance FID, we could explore the success of perceptual and GAN losses, as seen in VQGAN, ViT_VQGAN, and Stable Diffusion.
>
> |Method|Latent Size| Channel|logit-laplace loss|$L_2$ loss|Perceptual loss|GAN loss|FID&#8595;|PSNR&#8593;|
> |---|---|---|---|---|---|---|---|---|
> | DALL-E |16x16|--|&check;||||32.0|22.8|
> | VQGAN |16x16|256|||&check;|&check;|4.98|19.9|
> | ViT-VQGAN |32x32|32|&check;|&check;|&check;|&check;|1.28|--|
> |Stable Diffustion|16x16|16|||&check;|&check;|**0.87**|24.1|
> | **FINOLA (our)**|1x1|3072||&check;|||27.8|**25.8**|
>
> It's essential to clarify that FINOLA does not operate as a generative model. Our primary goal is not image generation; instead, we aim to unveil mathematical properties within the latent space. Image reconstruction, one of our tasks alongside self-supervised learning, serves as a means to validate FINOLA — a first-order norm+linear autoregressive model — as a fundamental mathematical property shared by all images.
>
> FINOLA is also implemented differently from autoregression-based generative models. The latter typically consists of two stages: the first stage involves learning an autoencoder and vector quantization in the latent space, while the second stage focuses on learning autoregression in the quantized latent space. In contrast, FINOLA operates as a single-stage model, seamlessly learning the encoder, autoregression, and decoder end-to-end, with autoregression performed on a continuous (non-quantized) latent space.

---

### Official Review · Reviewer_FEae · 2023-11-01

**Soundness:** 3 good
**Presentation:** 3 good
**Contribution:** 3 good
**Rating:** 5
**Confidence:** 3

**Summary:**

This paper introduces a novel framework FINOLA (First-Order Norm+Linear Autoregressive) for image representation with powerful autoregressive capabilities. First, image is firstly encoded into a single vector q. Then, FINOLA automatically regresses from the vector q placed in the center to the feature map through two partial differential equations. Experiments show that this pre-trained representation excels in various downstream tasks, including image classification and object detection, without the need for extensive fine-tuning.

**Strengths:**

1. The proposed method sounds interesting and novel. Its structure is simple but it shows powerful representation ability in spatial representation.
2. This paper provides a new perspective to describe the intrinsic relationship of image feature maps through partial differential equations. This may have some implications for simplifying neural networks.
3. Expensive experiments prove the effectiveness of this method, especially in image classification and object detection tasks.

**Weaknesses:**

1. The author mentioned "this intriguing property reveals a mathematical invariance". What does invariance refer to should be further explained.
2. The author mentioned "providing insights into the underlying mathematical principles" more than once, but did not provide an in-depth explanation or comparison. It is recommended to provide more details.
3. When validating the norm+linear approach, the authors repeated q by W × H times. In the original setting, the authors learned two matrices A and B, with more parameters. Thus, the comparison seems a bit unfair.
4. In comparable performance with Stable Diffusion, the input of the paper method is an image of the same size as the output, while the input of stable diffusion is a lower resolution image. The tasks of the two are different, so it may not be appropriate to compare them together.

**Questions:**

Please refer to the weakness part.

---

> ### Author Response · Authors · 2023-11-17
> **Authors' Response**
>
> Thank you for dedicating your time and effort to provide feedback on our work. Below, we answer the questions that have been raised.
>
> ---
> **Q1: The author mentioned "this intriguing property reveals a mathematical invariance". What does invariance refer to should be further explained.**
>
> The term "mathematical invariance" refers to the ***consistent relationship shared by all images between feature values and their spatial derivatives on the feature map***. Mathematically, for any given image, the feature values $\mathbfit{z}(x, y) \in \mathbb{R}^C$ at any position $(x, y)$ on the feature map determine its spatial derivatives as follows:
>
> $\mathbfit{z}(x + 1, y) - \mathbfit{z}(x, y) = \mathbfit{A}\mathbfit{z}_n(x, y)$,
>
> $\mathbfit{z}(x, y + 1) - \mathbfit{z}(x, y) = \mathbfit{B}\mathbfit{z}_n(x, y)$,
>
> where $\mathbfit{z}_n(x, y) = \frac{\mathbfit{z}(x, y) - \mu_z}{\sigma_z}$.
>
> It's essential to note that matrices $\mathbfit{A}$ and $\mathbfit{B}$ exhibit ***invariance*** across different images and pixels on the feature map, emphasizing the consistent mathematical relationship that holds universally.
>
> ---
> **Q2: The author mentioned "providing insights into the underlying mathematical principles" more than once, but did not provide an in-depth explanation or comparison. It is recommended to provide more details.**
>
> Thank you for the valuable suggestion. Allow us to provide a more detailed explanation. The crux of our mathematical insights lies in the understanding that the feature maps of all images adhere to two partial differential equations (PDEs):
>
> $\frac{\partial \mathbfit{z}}{\partial x}=\mathbfit{A}\mathbfit{z}_n, \quad \frac{\partial \mathbfit{z}}{\partial y}=\mathbfit{B}\mathbfit{z}_n$
>
> These equations reveal that the spatial rate of change, or derivatives, in the feature map is entirely determined by the feature values at the current location. It's essential to note that matrices $\mathbfit{A}$ and $\mathbfit{B}$ exhibit ***invariance*** across different images and pixels on the feature map, emphasizing the consistent mathematical relationship that holds universally.
>
> Another mathematical insight is related to the role of masking, particularly in the context of masked FINOLA for self-supervised learning. This technique plays a crucial role in facilitating the extraction of meaningful semantic representations from the feature maps. However, it comes with an inherent trade-off, impacting the preservation of intricate image details. Notably, compared to vanilla FINOLA, masked FINOLA exhibits a substantial increase in Gaussian curvature on the surfaces of critical features. This observed increase suggests a heightened curvature in the latent space, underlining its effectiveness in capturing and emphasizing semantic information.
>
> ---
> **Q3: When validating the norm+linear approach, the authors repeated q by W × H times. In the original setting, the authors learned two matrices A and B, with more parameters. Thus, the comparison seems a bit unfair.**
>
> The objective of this comparison is to demonstrate the superiority of using norm+linear over its *absence*. Repetition, requiring zero additional effort, naturally stands out as a straightforward option. In this context, our intention is to specifically exclude the scenario where the zero-effort repetition achieves performance comparable to our more complex norm+linear approach. Such a scenario would contradict the essential need for norm+linear. The poor performance of repetition serves as evidence that norm+linear cannot be easily replaced by a simpler and more straightforward solution, highlighting the necessity of our chosen approach, even in the presence of additional parameters when learning matrices $\mathbfit{A}$ and $\mathbfit{B}$.
>
> ---
> **Q4: In comparable performance with Stable Diffusion, the input of the paper method is an image of the same size as the output, while the input of stable diffusion is a lower resolution image. The tasks of the two are different, so it may not be appropriate to compare them together.**
>
> Let's clarify the comparison with Stable Diffusion. Our comparison is specifically with the ***first stage*** of Stable Diffusion, known as *perceptual image compression*, which involves training an autoencoder to reconstruct images. We explicitly refrain from comparing with the second stage of Stable Diffusion, which includes generation of super-resolution from lower-resolution images. The comparison is thus limited to the common task of image reconstruction and does not involve tasks related to super-resolution from lower-resolution inputs.

---

### Official Review · Reviewer_P6tB · 2023-11-01

**Soundness:** 3 good
**Presentation:** 3 good
**Contribution:** 2 fair
**Rating:** 6
**Confidence:** 3

**Summary:**

This paper proposed an auto-regression model in latent feature space using norm-linear transform to regress the features in high-resolution from a global feature. The proposed norm-linear transform is simple to regress features, and can be decoded with lightweight network to generate the reconstructed image. The proposed model can also be applied as a pre-training method, having good generalization ability to recognition and object detection.

**Strengths:**

1.  The auto-regression for generating features in high-resolution in feature space is an interesting idea, and the proposed regression model is simple. The regression model in feature space can be seen as a discretized PDE.

2. The proposed model can be taken as a self-supervised leaning approach based on mask region prediction. The sufficient experiments show that it can achieve good pretraining results for recognition and detection.

**Weaknesses:**

1.  There are previous regression-based generative models (e.g., refer to related works) in feature space, and what are the major difference and advantage of this approach compared with these models? Is it possible to compare with them for the generation quality and computational speed?

2. Are the matrix A and B shared for all different images and pixels in the feature space? If it is, why learned constant A, and B can deduce good feature regression?

3. The paper states that this work does not aim to achieve SoTA results, however, comparisons with SoTA regression models or other variants, e.g., using nonlinear regression instead of linear transform, should be able to better give insights to audience.

4. The regression model should gradually regression dense feature maps? How about the computational overhead/speed for training and inference using this model?

5. What is the meaning of the mathematical invariance in this paper for the proposed model?

**Questions:**

Please see my questions above.

---

> ### Author Response · Authors · 2023-11-17
> **Authors' Response (part 1)**
>
> Thank you for dedicating your time and effort to provide feedback on our work. Below, we answer the questions that have been raised.
>
> ---
> **Q1: There are previous regression-based generative models (e.g., refer to related works) in feature space, and what are the major difference and advantage of this approach compared with these models? Is it possible to compare with them for the generation quality and computational speed?**
>
> First and foremost, it's essential to clarify that FINOLA does *not* function as a generative model. Our primary objective is not image generation but rather the revelation of mathematical properties within the latent space. Image reconstruction, one of our tasks (alongside self-supervised learning), serves as a means to validate FINOLA — a first-order norm+linear autoregressive model — as a fundamental mathematical property shared by all images.
>
> It's crucial to note that FINOLA is both functionally and technically distinct from autoregression-based generative models. The latter typically consists of two stages: the first stage involves learning an autoencoder and vector quantization in the latent space, while the second stage focuses on learning autoregression in the quantized latent space. In contrast, FINOLA operates as a single-stage model, seamlessly learning the encoder, autoregression, and decoder end-to-end, with autoregression performed on a continuous (non-quantized) latent space.
>
> Given that FINOLA doesn't fall into the category of generative models, it isn't suitable for direct comparison in terms of generation quality and computational speed with models designed for such purposes.
>
> ---
> **Q2: Are the matrix A and B shared for all different images and pixels in the feature space? If it is, why learned constant A, and B can deduce good feature regression?**
>
> Yes, the matrices $\mathbfit{A}$ and $\mathbfit{B}$ are shared for all images and pixels in the feature space.
>
> The success of using constant $\mathbfit{A}$ and $\mathbfit{B}$ for achieving good feature regression can be attributed to three primary factors. Firstly, images inherently exhibit significant spatial redundancy, leading to strong correlations between consecutive positions within the feature map. Secondly, our hypothesis regarding the existence of a feature map where spatial derivatives correlate with feature values appears to be supported. Lastly, the effectiveness of deep learning allows us to validate this hypothesis by seamlessly learning the encoder, FINOLA, and decoder components in an end-to-end manner.
>
> ---
> **Q3: The paper states that this work does not aim to achieve SoTA results, however, comparisons with SoTA regression models or other variants, e.g., using nonlinear regression instead of linear transform, should be able to better give insights to audience.**
>
> Excellent point! We have conducted a comparison between our norm+linear approach and non-linear regression, which involves replacing the linear transform with two MLP layers incorporating GELU activation in between. This evaluation was performed for both 1024 and 3072 channels. Surprisingly, our norm+linear, despite having fewer parameters, closely trails the performance of the two-layer MLP. This result underscores the effectiveness of norm+linear in modeling the transition between pixels in the feature map.
>
> |Regression|C=1024|C=3072|
> |---|---|---|
> |Non-Linear (2 layer MLP)|23.8|25.8|
> |**Linear (our)**|23.7|25.8|
>
> We further benchmark our method against an alternative non-linear solution: employing a multi-layer convolutional approach to upsample the vector $\mathbfit{q}$ to a 16x16 feature map. This method undergoes four upsample stages, each incorporating three 3x3 convolution layers. The results below  demonstrate the superior performance of a single layer FINOLA over the multi-layer convolutional baseline.
>
> |Regression|C=1024|C=3072|
> |---|---|---|
> | convolution | 23.4 | 25.1 |
> | **FINOLA (our)** | **23.7** | **25.8** |

---

> > ### Author Response · Authors · 2023-11-17
> > **Authors' Response (part 2)**
> >
> > **Q4: The regression model should gradually regression dense feature maps? How about the computational overhead/speed for training and inference using this model?**
> >
> > ***Autoregressive reconstruction:*** The training of the autoregressive model involves regressing dense feature maps, and the computational requirements increase with the size of the feature map. For instance, training on a 16x16 feature map with 3072 latent channels for 100 epochs on ImageNet takes approximately 8 days with 8 V100 GPUs, and extending to a larger feature map like 64x64 increases the training time to 18 days using the same GPU setup.
> >
> > The complete inference pipeline, encompassing encoding, FINOLA, and decoding, for a 256x256 image on a MacBook Air with an Apple M2 CPU, takes approximately 1.2 seconds.
> >
> > ***Self-supervised learning:*** Training a self-supervised learning model (Masked-FINOLA-B) for 800 epochs on ImageNet demands around 10 days with 8 V100 GPUs and a batch size of 128 per GPU.
> >
> > It's worth noting that following self-supervised learning, only the encoder is typically used for inference in downstream tasks. Our method utilizes Mobile-Former as an encoder, which is a lightweight architecture efficiently combining MobileNet and Transformer.  The original paper *"Mobile-Former: Bridging MobileNet and Transformer, CVPR 2022"* shows detailed information on runtime performance (comparable with MobileNet and ShuffleNet).
> >
> > ---
> > **Q5: What is the meaning of the mathematical invariance in this paper for the proposed model?**
> >
> > The term "mathematical invariance" refers to the ***consistent relationship shared by all images between feature values and their spatial derivatives on the feature map***. Mathematically, for any given image, the feature values $\mathbfit{z}(x, y) \in \mathbb{R}^C$ at any position $(x, y)$ on the feature map determine its spatial derivatives as follows:
> >
> > $\mathbfit{z}(x + 1, y) - \mathbfit{z}(x, y) = \mathbfit{A}\mathbfit{z}_n(x, y)$,
> >
> > $\mathbfit{z}(x, y + 1) - \mathbfit{z}(x, y) = \mathbfit{B}\mathbfit{z}_n(x, y)$,
> >
> > where $\mathbfit{z}_n(x, y) = \frac{\mathbfit{z}(x, y) - \mu_z}{\sigma_z}$.
> >
> > It's essential to note that matrices $\mathbfit{A}$ and $\mathbfit{B}$ exhibit ***invariance*** across different images and pixels on the feature map, emphasizing the consistent mathematical relationship that holds universally.

---

### Official Review · Reviewer_JB6i · 2023-11-02

**Soundness:** 3 good
**Presentation:** 3 good
**Contribution:** 3 good
**Rating:** 6
**Confidence:** 3

**Summary:**

This paper is proposing First-Order Norm+Linear Autoregressive (FINOLA). The proposed method is a new type of autoregressive model that can be used for self-supervised representation learning. The comprehensive experiments show that FINOLA has relatively small parameters but can contain enough information for image reconstruction. The authors also show that FINOLA can be used as a feature extractor for the downstream task.

**Strengths:**

- Reasonable and understandable writing.
- technically novel.
- Comprehensive experiments.

**Weaknesses:**

- Justifications for the novel design choice (but seems heuristic) compared to the regular AR model.
    - Predefined assignments in Block-wise Masked FINOLA
    - Predicting three points in Block-wise Masked FINOLA
    - Why first order?
- Lack of interpretation of the derived PDE.
- Some parts are unclear (please see the Questions below)

**Questions:**

1. (page 2) The authors argued that “the coefficient matrices $A$ and $B$ capture the relationship between each position“. How do the coefficient matrices directly know the spatial relationship? They do not take any of the spatial information. I believe there are some pieces of missing information:
    1. coefficient matrices $A$ and $B$ capture channel-wise relationship.
    2. The pattern and correlation encoded in the channel are related to the positional information.
    3. the coefficient matrices $A$ and $B$ (indirectly) capture the relationship between each position.
Maybe it is trivial for some readers but it is not for me.

2. There are some of the questions about PDE.
    1. (page 4) “They represent a theoretical extension of FINOLA from a discrete grid to continuous coordinates.“ This is not intuitive to me. As far as I understand, the proposed method in this paper is also using the discrete grid. Assuming that the extension to continuous coordinates is to get a better theoretical understanding, what is the insight and take away from the fact that Eq. 1 becomes the formulation of PDE in Eq. 4?
    2. (page 4) “Establishing their theoretical validity poses a substantial challenge.”  What is the substantial challenge?
3. The specific method of the block-wise Masked FINOLA is unclear. For example, if we see the Corner case, let’s say an input coordinate is (3,2) and it is used for predicting {(11,10), (11,2), (3,10)}. I guess for obtaining (11,2), for instance, the function $\phi^8(z(3,2))$ is applied, which means (4,2) is predicted first and it is used as an input for predicting (5,2) and so forth. My question is, considering that we already have a ground truth within (0,2),(1,2) …, (7,2), why not use the ground truth information?
4. How is the Gaussian curvature related to capturing semantics? Could you add more detailed descriptions of how it is computed?
5. (Table 4 (a)) Even though Stable Diffusion is a Generative Model, I believe it should have better PSNR than FINOLA for the image reconstruction task. How did you implement the image reconstruction task for Stable Diffusion?
6. How fast is the parallel implementation (Fig. 3) compared to the regular AR setting?
7. Is this method fast enough to use as a feature extractor for the downstream task?

---

> ### Author Response · Authors · 2023-11-17
> **Authors' Response (part 1)**
>
> Thank you for dedicating your time and effort to provide feedback on our work. Below, we answer the questions that have been raised.
>
> ---
> **Weakness 1: Justifications for the novel design choice (but seems heuristic) compared to the regular AR model.**
>
> Thanks for the suggestion. Below we provide justification for each item.
>
> ***Predefined assignments and predicting three points in Block-wise Masked FINOLA***
>
> This design is meticulously crafted to achieve several key objectives: (a) Full coverage: ensuring that all masked positions receive attention, (b) Balance: distributing the usage of unmasked positions equally, and (c) Efficiency: utilizing each unmasked position to predict three masked positions, especially under the condition where 75% of positions are masked. Additionally, this design can be implemented in a block-wise manner to further enhance efficiency, a crucial consideration for the extended training durations involved in self-supervised learning.
>
> ***Why first order?***
>
> The selection of first-order autoregression is driven by its inherent simplicity and local property, where the derivatives along the $x$ and $y$ axes are exclusively determined by the current position. This strategic choice not only simplifies the computational process but also aims to unveil valuable mathematical insights that could be shared across different images, provided the approach proves effective."
>
> ---
> **Weakness 2: Lack of interpretation of the derived PDE.**
>
> The derived partial differential equations (PDEs) in Eq. 4 provide a theoretical extension of FINOLA from a discrete grid to continuous coordinates, establishing a conceptual bridge between the discrete representations of neural networks and the continuous nature of human vision. Although our digital images and feature maps are inherently discrete, the human vision system perceives continuous signals.
>
> This theoretical extension allows us to hypothesize about the mathematical properties of continuous feature representations in the human vision system. We posit that the proposed PDEs in Eq. 4, derived from their discrete counterparts in Eq. 1, capture the patterns inherent in continuous features within human vision.
>
> This is related to **Q2** below.
>
> ---
> **Q1: How do the coefficient matrices $\mathbfit{A}$ and $\mathbfit{B}$ directly know the spatial relationship? They do not take any of the spatial information.**
>
> The coefficient matrices $\mathbfit{A}$ and $\mathbfit{B}$ encode the relationship between a ***feature vector*** $\mathbfit{z}(x, y) \in \mathbb{R}^C$ and and ***its spatial derivatives***, namely $\mathbfit{z}(x+1, y)-\mathbfit{z}(x, y)$ and $\mathbfit{z}(x, y+1)-\mathbfit{z}(x, y)$. This relationship is mathematically expressed as:
>
> $\mathbfit{z}(x + 1, y) - \mathbfit{z}(x, y) = \mathbfit{A}\mathbfit{z}_n(x, y)$,
>
> $\mathbfit{z}(x, y + 1) - \mathbfit{z}(x, y) = \mathbfit{B}\mathbfit{z}_n(x, y)$,
>
> where $\mathbfit{z}_n(x, y) = \frac{\mathbfit{z}(x, y) - \mu_z}{\sigma_z}$.
>
> It's essential to note that matrices $\mathbfit{A}$ and $\mathbfit{B}$ exhibit invariance across different images and positions $(x,y)$ within the feature map once they are learned from the data.
>
> ***1.1 Coefficient matrices $\mathbfit{A}$ and $\mathbfit{B}$ capture channel-wise relationship.***
>
> Indeed, the difference of the channel vector between consecutive positions, such as $\mathbfit{z}(x + 1, y) - \mathbfit{z}(x, y)$, represents a linear combination of normalized channel values at the current position $\mathbfit{z}_n(x, y) \in \mathbb{R}^C$.
>
> ***1.2: The pattern and correlation encoded in the channel are related to the positional information.***
>
> The correlation does *NOT* pertain to the positional information; instead, it is associated with the derivatives (or rate of change) along the $x$ and $y$ axes, often referred to as *"spatial derivatives"*.
>
> ***1.3: the coefficient matrices $\mathbfit{A}$ and $\mathbfit{B}$ (indirectly) capture the relationship between each position. Maybe it is trivial for some readers but it is not for me.***
>
> As discussed earlier, the matrices $\mathbfit{A}$ and $\mathbfit{B}$ explicitly capture the relationship between the feature at a given position $\mathbfit{z}(x, y)$ and its derivatives along the $x$ and $y$ axes. Consequently, this encoding implicitly extends to capturing the relationship between consecutive positions, such as from $\mathbfit{z}(x, y)$ to $\mathbfit{z}(x + 1, y)$.

---

> > ### Author Response · Authors · 2023-11-17
> > **Authors' Response (part 2)**
> >
> > **Q2: There are some of the questions about PDE.**
> >
> > ***2.1 (page 4) “They represent a theoretical extension of FINOLA from a discrete grid to continuous coordinates.“ This is not intuitive to me. As far as I understand, the proposed method in this paper is also using the discrete grid. Assuming that the extension to continuous coordinates is to get a better theoretical understanding, what is the insight and take away from the fact that Eq. 1 becomes the formulation of PDE in Eq. 4?***
> >
> > The mention of extending from a discrete grid to continuous coordinates in Eq. 4 represents a theoretical consideration. While our digital images and their feature maps are inherently discrete, we draw a connection to the human vision system, which perceives continuous signals. The extension to continuous coordinates in Eq. 4 is more of a conceptual bridge to relate our discrete representation to the continuous nature of human vision.
> >
> > This theoretical extension allows us to postulate the mathematical characteristics of continuous feature representations within the human visual system. We posit that the PDEs proposed in Eq. 4, evolving from their discrete counterparts in Eq. 1, effectively encapsulate the inherent patterns of continuous features in human vision.
> >
> > Furthermore, employing PDEs serves as a guide for validation. If these PDEs hold true, we should be able to leverage them to generate features at multiple resolutions. Our experimental results indeed validate this hypothesis, demonstrating that FINOLA consistently performs well across various resolutions, ranging from 8×8 to 256×256. Particularly noteworthy is the performance at the highest resolution (256x256), where the feature map matches the image size, and the decoder remains lightweight with only three 3x3 convolutional layers. This showcases the effectiveness of our approach.
> >
> > ***2.2: (page 4) “Establishing their theoretical validity poses a substantial challenge.” What is the substantial challenge?***
> >
> > The substantial challenge in establishing the theoretical validity of the partial differential equations (PDEs) over the feature map on continuous coordinates lies in the inaccessibility of the feature space of the human vision system. The theoretical validation involves hypothesizing that these PDEs reveal the mathematical properties of continuous feature representations in the human vision system.
> >
> > However, the direct validation of these hypotheses on the feature space of the human vision system is impractical due to its inaccessibility. Unlike the feature maps generated by neural networks, which are discrete and accessible, the true continuous feature space in human vision remains beyond our reach for direct theoretical validation. Therefore, we resort to empirical validation on the discrete grid of feature maps generated by neural networks as a practical and accessible means to support our hypotheses and demonstrate the effectiveness of our approach.

---

> > > ### Author Response · Authors · 2023-11-17
> > > **Authors' Response (part 3)**
> > >
> > > **Q3: The specific method of the block-wise Masked FINOLA is unclear. For example, if we see the Corner case, let’s say an input coordinate is (3,2) and it is used for predicting {(11,10), (11,2), (3,10)}. I guess for obtaining (11,2), for instance, the function is applied, which means (4,2) is predicted first and it is used as an input for predicting (5,2) and so forth. My question is, considering that we already have a ground truth within (0,2),(1,2) …, (7,2), why not use the ground truth information?**
> > >
> > > Great observation! The current design of block-wise Masked FINOLA intentionally avoids utilizing available ground truth along the predicting path (e.g., excluding (0,2), (1,2), ..., (7,2)). This decision stems from careful considerations related to both *training speed* and the *difficulty of the prediction task*.
> > >
> > > ***Training speed:*** Employing multiple ground truths in the unmasked block to predict every masked position can be slow. In our current solution, each masked position is predicted from a single unmasked position, enhancing efficiency. Furthermore, the prediction process can be implemented in parallel at the group level. For instance, applying horizontal FINOLA eight times on the whole block (0:8, 0:8) to predict the block on the right (8:16, 0:8) is possible because all unmasked positions within a group (0:8, 0:8) share offsets towards their assigned masked positions. This parallelism is crucial for self-supervised pre-training, which requires an extended training period (800 or 1600 epochs). However, this advantage is not applicable when involving multiple ground truths to predict every masked position.
> > >
> > > ***Difficulty of prediction:*** Using closer ground truth (e.g., predicting (11,2) from (7,2) rather than from (3,2)) reduces the prediction challenge, potentially compromising representation learning. We conducted an experiment by constraining the prediction offset to less than 4. For instance, if the unmasked block is (0:8, 0:8), the final feature map is (0:12, 0:12) rather than (0:16, 0:16). Specifically, we used (4:8, 0:8) to predict (8:12, 0:8), (0:8, 4:8) to predict (0:8, 8:12), and (4:8, 4:8) to predict (8:12, 8:12). The linear probing performance dropped from 66.4 to 63.9.
> > >
> > > The design of block-wise Masked FINOLA carefully balances considerations of training speed and pre-training difficulty. We appreciate your feedback and will enhance the clarity of these details in the final draft.
> > >
> > > ---
> > > **Q4: How is the Gaussian curvature related to capturing semantics? Could you add more detailed descriptions of how it is computed?**
> > >
> > > ***Calculation of Gaussian curvature:*** The Gaussian curvature is computed per feature channel. Consider the $k^{th}$ channel, where its feature map $z_k(x,y)$ represents a 2D surface in the $(x, y, z)$ 3D space. At each position $(x, y)$, the Gaussian curvature is determined as follows:
> > >
> > > $\kappa_k(x,y) = \frac{\frac{\partial^2 z_k}{\partial x^2}\frac{\partial^2 z_k}{\partial y^2}-\left( \frac{\partial^2 z_k}{\partial x \partial y}\right)^2}{\left(1+(\frac{\partial z_k}{\partial x})^2+(\frac{\partial z_k}{\partial y})^2\right)^2}$.
> > >
> > > Finite differences are employed to approximate the first and second-order derivatives, resulting in a curvature map. The root mean square of the peak positive curvature ($\kappa_{k+}$) and the peak negative curvature ($\kappa_{k-}$) per channel is then computed as:
> > >
> > > $\kappa_k^*=\sqrt{(\kappa_{k+}^2+\kappa_{k-}^2)/2}$
> > >
> > > Channels are ranked based on $\kappa_k^*$, focusing on the top-K channels with significantly curved feature surfaces $z_k(x,y)$.
> > >
> > > ***Relation to capturing semantics:*** The correlation between Gaussian curvature and capturing semantics is observed through experiments. Firstly, Masked FINOLA (pre-trained with masking) exhibits significantly higher image classification performance than vanilla FINOLA (66.4 vs. 17.9 on linear probing) but performs significantly worse on image reconstruction (17.3 vs. 25.8 PSNR), indicating that the Masked FINOLA pre-trained encoder trades off details for capturing more semantics. Secondly, Masked FINOLA demonstrates significantly larger Gaussian curvature $\kappa_k^*$ on the top-K channels compared to vanilla FINOLA. This correlation suggests a link between semantics and larger Gaussian curvature on critical features. Further exploration of this correlation is left for future work.

---

> > > > ### Author Response · Authors · 2023-11-17
> > > > **Authors' Response (part 4)**
> > > >
> > > > **Q5: (Table 4 (a)) Even though Stable Diffusion is a Generative Model, I believe it should have better PSNR than FINOLA for the image reconstruction task. How did you implement the image reconstruction task for Stable Diffusion?**
> > > >
> > > > It's important to note that even though Stable Diffusion is a generative model, the comparison in this context was limited to the performance of the ***first stage***, referred to as *perceptual image compression*. In this stage, an autoencoder is trained to reconstruct images. The performance metrics for Stable Diffusion were obtained from the original paper titled *"High-Resolution Image Synthesis with Latent Diffusion Models, CVPR 2022,"* specifically from Table 8 in Appendix D. The latent size chosen for this comparison was 16x16 ($f=16$).
> > > >
> > > > ---
> > > > **Q6: How fast is the parallel implementation (Fig. 3) compared to the regular AR setting?**
> > > >
> > > > The table below presents a comparison between our parallel implementation and the regular autoregressive (AR) setting in terms of inference runtime. The runtime evaluation considers the complete inference pipeline, including encoding, autoregression, and decoding, for a 256x256 image on a MacBook Air with an Apple M2 CPU. The comparison is conducted for two feature map sizes, 16x16 and 64x64.
> > > >
> > > > |AR|16x16|64x64|
> > > > |---|---|---|
> > > > |Regular AR|1.7s|7.7s|
> > > > |**Parallel (our)**|**1.2s**|**2.6s**|
> > > >
> > > > The parallel implementation demonstrates superior speed compared to the regular AR setting. It achieves a 30% speedup at a resolution of 16x16 and a threefold increase in speed at a higher resolution of 64x64.
> > > >
> > > > ---
> > > > **Q7: Is this method fast enough to use as a feature extractor for the downstream task?**
> > > >
> > > > Yes, our method utilizes Mobile-Former as an encoder, which is a lightweight architecture efficiently combining MobileNet and Transformer.  The original paper "*Mobile-Former: Bridging MobileNet and Transformer, CVPR 2022*" shows detailed information on runtime performance (comparable with MobileNet and ShuffleNet). This paper demonstrates its effective performance for downstream tasks like image classification and object detection, after FINOLA pre-training.

---

### Author Response · Authors · 2023-11-20
**Reminder: Approaching End of Author-Reviewer Discussion Period**

Dear Reviewers,

We hope this message finds you well. We want to extend a friendly reminder that the author-reviewer discussion period is nearing its close. Your input and perspective have been highly valuable throughout this process. If any questions or concerns about our rebuttal arise, we're here to listen and address them in a timely manner. Your insights are crucial in refining our manuscript further.

Understanding the demands on your time, we deeply appreciate your active participation in this rebuttal discussion. Your contributions are invaluable to us, and we genuinely thank you for your dedication.

Should you have any inquiries or thoughts, please don't hesitate to reach out.

Best regards,

-authors

---

> ### Author Response · Authors · 2023-11-21
> **Second Reminder: Two Days Remaining for Author-Reviewer Discussion Period**
>
> Dear Reviewers,
>
> We are writing to remind you that the author-reviewer discussion period is soon coming to an end. If you have any concerns or questions regarding our rebuttal, please do not hesitate to inform us. Your input will be valuable in helping us refine our manuscript.
>
> Understanding the demands on your schedule, we sincerely appreciate the time you've dedicated to this rebuttal discussion. Your thoughtful feedback is immensely helpful to us.
>
> Thank you for your attention.
>
> Best regards,
>
> -authors

---

### Meta-Review · Area_Chair_upTT · 2023-12-05

**Metareview:**

This paper introduces FINOLA, a novel framework that represents each image as a first-order norm+linear autoregressive process, revealing the presence of underlying partial differential equations (PDEs) governing the latent feature space. The authors validate the proposed method through experiments in image reconstruction and self-supervised learning. While the method itself is interesting and novel, improvements are needed in writing and narrative clarity. Some details and necessary explanations are omitted, causing confusion. The rebuttals partially addressed concerns raised by reviewers. However, during internal discussions, all reviewers unanimously agree to reject the paper due to unconvincing experimental results (Reviewer QCbo and Reviewer P6tB), lack of comparison with state-of-the-art regression-based models (Reviewer P6tB), and unclear statements and claims (Reviewer P6tB and Reviewer JB6i). Considering these factors, the Area Chair recommends rejecting the paper in its current state. The Area Chair encourages the authors to enhance their paper by carefully considering all the valuable suggestions provided by the reviewers, aiming for submission to the next venue.

**Justification For Why Not Higher Score:**

All the reviewers reach a consensus to reject the paper due to the following concerns, such as unconvincing experimental results (Reviewer QCbo and Reviewer P6tB), lack of comparison with state-of-the-art regression-based models (Reviewer P6tB), and unclear statements and claims (Reviewer P6tB and Reviewer JB6i). The paper is not ready for publication.

**Justification For Why Not Lower Score:**

NA

---

### Decision · Program_Chairs · 2024-01-16

Reject